# BEYOND WORST-CASE ATTACKS: ROBUST RL WITH ADAPTIVE DEFENSE VIA NON-DOMINATED POLICIES

**Xiangyu Liu**[†1]**, Chenghao Deng**[†1]**, Yanchao Sun**[2]**, Yongyuan Liang**[1]**, Furong Huang**[1]
[1]University of Maryland, College Park, [2]J.P. Morgan AI Research
{xyliu999}@umd.edu

## ABSTRACT

In light of the burgeoning success of reinforcement learning (RL) in diverse real-world applications, considerable focus has been directed towards ensuring RL policies are robust to adversarial attacks during test time. Current approaches largely revolve around solving a minimax problem to prepare for potential worst-case scenarios. While effective against strong attacks, these methods often compromise performance in the absence of attacks or the presence of only weak attacks. To address this, we study policy robustness under the well-accepted state-adversarial attack model, extending our focus beyond only worst-case attacks. We first formalize this task at test time as a regret minimization problem and establish its intrinsic hardness in achieving sublinear regret when the baseline policy is from a general continuous policy class, $\Pi$. This finding prompts us to *refine* the baseline policy class $\Pi$ prior to test time, aiming for efficient adaptation within a finite policy class $\widetilde{\Pi}$, which can resort to an adversarial bandit subroutine. In light of the importance of a small, finite $\widetilde{\Pi}$, we propose a novel training-time algorithm to iteratively discover *non-dominated policies*, forming a near-optimal and minimal $\widetilde{\Pi}$, thereby ensuring both robustness and test-time efficiency. Empirical validation on the Mujoco corroborates the superiority of our approach in terms of natural and robust performance, as well as adaptability to various attack scenarios.

## 1 INTRODUCTION

With an increasing surge of successful applications powered by reinforcement learning (RL) on robotics (Levine et al., 2016; Ibarz et al., 2021), creative generation (OpenAI, 2023), etc, its safety issue has drawn more and more attention. There has been a series of works devoted to both the attack and defense aspects of RL (Kos & Song, 2017; Huang et al., 2017; Pinto et al., 2017; Lin et al., 2019b; Tessler et al., 2019; Gleave et al., 2019). Specifically, the vulnerability of RL policies has been revealed under various strong threats, which in turn facilitates the training of deep RL policies by accounting for the potential attacks to boost the robustness.

Existing approaches aimed at principled defense often prioritize robustness against worst-case attacks (Tessler et al., 2019; Russo & Proutiere, 2019; Zhang et al., 2021; Sun et al., 2021; Liang et al., 2022), focusing on the optimal attacker policy within a potentially constrained attacker policy space. Such a focus can lead to suboptimal performance when RL policies are subjected to no or weak attacks during test time. Real-world scenarios often diverge from these worst-case assumptions for several reasons: (1) Launching an attack against an RL policy might first require bypassing well-protected sensors, thus constraining the attack's occurrence in terms of time steps and its intensity. (2) Previous studies (Zhang et al., 2021; Sun et al., 2021) have highlighted the intriguing difficulty of learning the optimal attack policy, particularly when attackers are constrained by algorithmic efficiency or computational resources. Given these practical considerations and the prevalence of non-worst-case attacks, we pose and endeavor to answer the following question:

*Is it possible to develop a comprehensive framework that enhances the performance of the victim against non-worst-case attacks, while maintaining robustness against worst-case scenarios?*

---

[†]Equal contribution. Codes are available at https://github.com/umd-huang-lab/PROTECTED.git

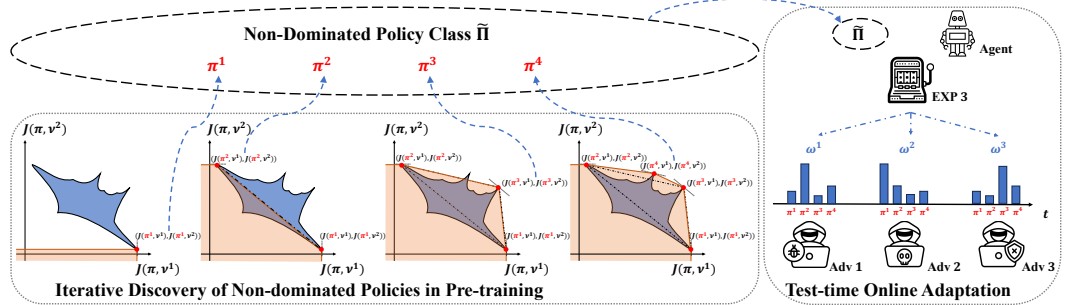

Figure 1: **Diagram of our** `PROTECTED` **framework. During training**, we iteratively discover non-dominated policies, forming a finite policy class $\widetilde{\Pi}$. The blue area delineates the reward landscape for victims against attackers, denoted as $\{(J(\pi, \nu^1), J(\pi, \nu^2)) \,|\, \pi \in \Pi\}$. Here, only two attackers are visualized for clarity. The orange area, on the other hand, represents the space of policies that are "*dominated*" by the discovered policy class $\widetilde{\Pi}$. Dominated policies are those that are outperformed by at least one (mixed) policy in $\widetilde{\Pi}$ across the specified range of attackers. We refer to §C for more detailed explanations. **During test time**, online adaptation mechanisms are activated to adjust the weight of each policy in response to various attack scenarios adaptively.

To address these challenges, we introduce `PROTECTED`, which stands for *pre-training non-dominated policies towards online adaptation*. In this work, the terms 'pre-training' and 'training' are used interchangeably. At test time, rather than deploying a single static policy, `PROTECTED` maintains a set of policies, denoted as $\widetilde{\Pi}$, and adaptively updates the weight of each policy based on interactions with the attacker to minimize regret. Before that, during training, a finite $\widetilde{\Pi}$ is constructed by iteratively discovering *non-dominated policies*, ensuring optimal defense against unknown attackers. In summary, our contributions encompass both training and online adaptation phases under the prevailing state-adversarial attack model:

▷ **(1) Online adaptation.** We formalize the problem of online adaptation and introduce regret minimization as the objective. We also highlight the inherent difficulty in achieving sublinear regret, advocating for a refined policy class $\widetilde{\Pi}$ for online adaptation.

▷ **(2) Non-dominated policy discovery during training.** For training, we characterize the optimality of $\widetilde{\Pi}$ and propose an algorithm for iteratively discovering non-dominated policies. This results in a $\widetilde{\Pi}$ that is both optimal and efficient for online adaptation, subject to certain mild conditions. Meanwhile, we also reveal the fundamental hardness of our problem that there are problem instances requiring a relatively large $\widetilde{\Pi}$ to achieve near-optimality.

▷ **(3) Empirical investigations.** Through empirical studies on Mujoco, we validate the effectiveness of `PROTECTED`, demonstrating both improved natural performance and robustness, as well as adaptability against unknown and dynamic attacks.

By investigating defenses against attacks beyond worst cases, we hope this work paves the way for the development of more practical defense mechanisms against a broader range of attack scenarios.

## 2 RELATED WORKS

**(State-)adversarial attacks on deep RL.** Early research by Huang et al. (2017) exposed the vulnerabilities of neural policies by adapting adversarial attacks from supervised learning to RL. Lin et al. (2019b) focused on efficient attacks, perturbing agents only at specific time steps. Following these works, there have been advancements in stronger pixel-based attacks (Qiaoben et al., 2021; Pattanaik et al., 2017; Oikarinen et al., 2020). Zhang et al. (2020a) introducing the theoretical framework SA-MDP for state adversarial perturbations and suggesting a corresponding regularizer for more robust RL policies. Building upon these, Sun et al. (2021) refined the framework to PA-MDP for improved efficiency. Liang et al. (2022) further improved the efficiency of defense by introducing the worse-case Q function, avoiding the alternating training. Those works as mentioned before aim at improving the robustness against worst-case attacks. Havens et al. (2018) also dealt

with the adversarial attacks for RL in an online setting, where it focuses on how to ensure robustness in the presence of attackers during RL training time.

**Online learning and meta-learning.** During the test phase, our framework equips the victim with the capability to adjust its policy in response to an unknown or dynamically changing attacker. This is achieved through the utilization of feedback from previous interactions. In the literature, two distinct paradigms have been advanced to examine how an agent can leverage historical tasks or experiences to inform future learning endeavors. The first paradigm, known as meta-learning (Schmidhuber, 1987; Vinyals et al., 2016; Finn et al., 2017), conceptualizes this as the task of "learning to learn." In meta-learning, prior experiences contribute to the formulation of a prior distribution over model parameters or instruct the optimization of a learning procedure. Typically, in this framework, a collection of meta-training tasks is made available together upfront. There are also works extending meta-learning to deal with the streaming of sequential tasks (Finn et al., 2019), which however require a task-specific update subroutine. The second paradigm falls under the rubric of online learning (Hannan, 1957; Cesa-Bianchi & Lugosi, 2006), wherein tasks—or in the context of our paper, attackers—are disclosed to the victim sequentially via bandit feedback. Extensive literature has been devoted to the subject of online learning, targeting the minimization of regret in either stochastic settings (Lattimore & Szepesvári, 2020; Auer, 2002; Russo & Van Roy, 2016) or adversarial settings (Auer et al., 2002; Neu, 2015; Jin et al., 2020). Our work primarily aligns with the latter paradigm. However, existing methodologies within this domain generally permit only reward functions to change arbitrarily, which is called the adversarial bandit problem or adversarial MDP problem. In contrast, our scenario permits the attacker to introduce partial observability for the victim, thereby also influencing the transition dynamics *from the perspective of the victim*.

## 3 PRELIMINARIES

In this section, we adopt the similar setup and notations as existing works (Zhang et al., 2021; Sun et al., 2021; Liang et al., 2022).

**MDP and attacker model.** We define a Markov decision process (MDP) as $\mathcal{M} = (\mathcal{S}, \mathcal{A}, \mathbb{T}, \mu_1, r, H)$, where $\mathcal{S}$ is the state space, $\mathcal{A}$ is the action space, $\mathbb{T} : \mathcal{S} \times \mathcal{A} \to \Delta(\mathcal{S})$ denotes the transition kernel, $\mu_1 \in \Delta(\mathcal{S})$ is the initial state distribution, $r_h : \mathcal{S} \times \mathcal{A} \to [0, 1]$ is the reward function for each $h \in [H]$. Given an MDP $\mathcal{M}$, at each step $h$, the attacker sees the true state $s_h \in \mathcal{S}$ and selects a perturbed state $\widehat{s}_h \in \mathcal{S}$ in a potentially adversarial way. Then the victim only sees the *perturbed* state $\widehat{s}_h$ instead of the true $s_h$ and takes the corresponding action $a_h \in \mathcal{A}$. The goal of the victim is to maximize its expected return while the attacker attempts to minimize it.

**Policy and value function.** We define the deterministic attacker policy $\nu = \{\nu_h\}_{h \in [H]}$ with $\nu_h : \mathcal{S} \to \mathcal{S}$ for any $h \in [H]$, and denote the corresponding policy space as $\mathcal{V}^{\mathrm{det}}$. We also consider constraints on the attacker, where for any $s$, the attacker can only perturb $s$ to some $\widehat{s} \in \mathcal{B}(s) \subseteq \mathcal{S}$, e.g., $\mathcal{B}(s)$ can be the $l_p$ ball. We allow randomized policies for the attacker and the policy space is denoted as $\mathcal{V} := \Delta(\mathcal{V}^{\mathrm{det}})$. For any $\nu \in \mathcal{V}$, we adopt the representation that $\nu$ is conditioned on a random seed $z \in \mathcal{Z}$ sampled at the beginning of each episode from a fixed probability distribution $\mathbb{P}(z)$. For the victim, we denote history $\tau_h$ at time $h$ as $\{\widehat{s}_1, a_1, \widehat{s}_2, a_2, \cdots, \widehat{s}_h\}$ and $\mathcal{T}$ as the space for all possible history at all steps. We consider history-dependent victim policy $\pi : \mathcal{T} \to \Delta(\mathcal{A})$ and $\Pi$ as the corresponding policy space. Finally, we use $\Pi^{\mathrm{det}}$ to denote deterministic victim policies. Given the victim policy $\pi$ and attacker policy $\nu$, the value function for the victim is defined as:
$J(\pi, \nu) = \mathbb{E}_{z \sim \mathbb{P}(z)} \mathbb{E}_{s_h \sim \mathbb{T}(\cdot \mid s_{h-1}, a_{h-1}), \widehat{s}_h \sim \nu_h(\cdot \mid s_h, z), a_h \sim \pi(\cdot \mid \tau_h)} [\sum_{h=1}^{H} r_h(s_h, a_h)]$.

## 4 THE PROTECTED FRAMEWORK

### 4.1 ONLINE ADAPTATION FOR ADAPTIVE DEFENSES

Before delving into our approach of online adaptation for adaptive defenses, it is essential to review the limitations of existing works concerning the trade-off between natural rewards and robustness. Then we discuss the necessity of an adaptive defending policy. Existing researches generally focus on worst-case performance, formally characterized as follows:

**Definition 4.1** (Exploitability). *Given a victim policy $\pi$, exploitability is defined by:*
$$\mathrm{Expl}(\pi) = \max_{\pi' \in \Pi} \min_{\nu \in \mathcal{V}} J(\pi', \nu) - \min_{\nu \in \mathcal{V}} J(\pi, \nu).$$

Existing works aim to obtain a policy $\pi^\star$ that minimizes exploitability, i.e., $\pi^\star \in \arg\min_\pi \mathrm{Expl}(\pi)$, during the training phase to defend against worst-case or strongest attacks. Such a trained policy, $\pi^\star$, is then deployed universally at test time. However, this approach can be overly cautious, compromising performance under no or weak attacks (Zhang et al., 2021; Sun et al., 2021). To address this limitation, we propose to consider a new metric for test-time performance:

**Definition 4.2** (Regret). *Given $T$ total episodes at test time, at the start of each episode $t$, the victim selects a policy $\pi^t$ from $\Pi$ based on reward feedback from previous episodes, and the attacker selects an arbitrary policy $\nu^t \in \mathcal{V}$. The regret is defined as* [1]

$$\mathrm{Regret}(T) = \max_{\pi \in \Pi} \sum_{t=1}^{T} \left( J(\pi, \nu^t) - J(\pi^t, \nu^t) \right), \tag{4.1}$$

Therefore, instead of employing a static victim policy, $\pi^\star$, designed to minimize exploitability, we propose adaptively selecting $\{\pi^t\}_{t \in [T]}$ during test time, based on online reward feedback, to minimize regret. Once the adaptively selected victims, $\{\pi^t\}_{t \in [T]}$, ensure low regret, the performance against either strong or weak (or even no) attacks is guaranteed to be near-optimal. While such an objective could ideally provide a way to defend against non-worst-case attacks, unfortunately, it turns out that there are no efficient algorithms that can *always* guarantee sublinear regret.

**Proposition 4.3.** *Fix $\alpha \in [0, 1)$. There does not exist an algorithm that produces a sequence of victim policies $\{\pi^t\}_{t \in [T]}$ such that $\mathbb{E}[\mathrm{Regret}(T)] = \mathrm{poly}(S, A, H)T^\alpha$ for any $\{v^t\}_{t \in [T]}$.*

**Remark 4.4.** *On the downside, Proposition 4.3 remains valid even when the attacker's actions are constrained such that $|\mathcal{B}(s)| = 2$ and $s \in \mathcal{B}(s)$ for each $s \in \mathcal{S}$. However, there is a silver lining: in the hard instance we constructed, the attacker must perturb a state $s$ to another state $\widehat{s}$ such that both the transition dynamics and the reward function differ greatly between $s$ and $\widehat{s}$. Therefore, if real-world scenarios impose constraints – such as $\|s - \widehat{s}\| \le \epsilon$ for some $\epsilon$ in continuous control tasks, and if the transition dynamics and reward function are locally Lipschitz – Proposition 4.3 may not apply. Further investigation of this avenue is left for future work.*

The earlier negative results inform us to focus on online adaptation within a smaller, finite policy class $\widetilde{\Pi}$, rather than the broader class $\Pi$. Specifically, in Equation 4.1, $\{\pi^t\}_{t \in [T]}$ and the best policy $\pi$ in hindsight in Definition 4.2 belongs to a rather large general policy class $\Pi$. Therefore, by relaxing the regret definition to ensure the baseline policy $\pi$ to come from a smaller and finite policy class $\widetilde{\Pi} \subseteq \Pi$, achieving sublinear regret becomes possible. This can be done by treating each policy in $\widetilde{\Pi}$ as one arm and running an adversarial bandit algorithm, e.g., EXP3 (Bubeck et al., 2012). Meanwhile, it is worth noting that if the test-time attacker is unknown but fixed, stochastic bandit algorithms like UCB can be also effective. Given such a refined policy class $\widetilde{\Pi}$, we can perform online adaptation as in Algorithm 1, which maintains a meta-policy $\omega^t \in \Delta(\widetilde{\Pi})$ during online adaptation and adjusts the weight of each policy based on the online reward feedback. The key is that the victim should randomize its policy by sampling from $\widetilde{\Pi}$, following the distribution $\omega^t$ at the start of each episode $t$. Formally, such an algorithm ensures the guarantees for a relaxed definition of regret, following the analysis of EXP3.

**Proposition 4.5** (Bubeck et al. (2012)). *Given $\widetilde{\Pi} \subseteq \Pi$ with $|\widetilde{\Pi}| < \infty$, we define $\widetilde{\mathrm{Regret}}(T) = \max_{\pi \in \widetilde{\Pi}} \sum_{t=1}^{T} (J(\pi, \nu^t) - J(\pi^t, \nu^t))$ for any $T \in \mathbb{N}$, $\{\pi^t\}_{t \in [T]}$, $\{\nu^t\}_{t \in [T]}$. Algorithm 1 for producing $\{\pi^t\}_{t \in [T]}$ enjoys the guarantees $\mathbb{E}[\widetilde{\mathrm{Regret}}(T)]/T \le 2H\sqrt{\frac{|\widetilde{\Pi}| \log |\widetilde{\Pi}|}{T}}$.*

Finally, we remark that the adaptation method used here is computationally efficient as it only maintains and updates the vector $\omega^t \in \mathbb{R}^{|\widetilde{\Pi}|}$, rather than fine-tuning a policy network (or its last layer). This makes it more suitable for scenarios where computational budgets are limited at test time.

---

[1] Regret depends on the specific attackers $\{\nu^t\}_{t \in [T]}$. However, we omit such dependency in Regret$(T)$ for notational convenience since both the regret upper bound of interest for our paper and the literature of online learning and adversarial bandit (Hazan et al., 2016; Lattimore & Szepesvári, 2020) will be for any $\{\nu^t\}_{t \in [T]}$.

---

**Algorithm 1** Online adaptation with refined policy class

---

**Input:** $\widetilde{\Pi}, T, \eta$
Initialize $\omega^1 \in \Delta(\widetilde{\Pi})$ to be the uniformly random distribution.
**for** $t \in [T]$ **do**
    Draw $\pi^t \sim \omega^t$                                            ▷ sampling randomly
    Execute $\pi^t$ in the underlying environment and observe total rewards $R^t(\pi^t) := \sum_{h=1}^H r_h$
    **for** $\pi \in \widetilde{\Pi}$ **do**
        $\omega^{t+1}(\pi) \leftarrow \frac{e^{\eta \sum_{s=1}^t \widehat{R}^s(\pi)}}{\sum_{\pi' \in \widetilde{\Pi}} e^{\eta \sum_{s=1}^t \widehat{R}^s(\pi')}}$, where $\widehat{R}^s(\pi) = \frac{R^s(\pi)}{\omega^s(\pi)} \mathbb{1}_{\pi = \pi^s}$ for $s \in [t]$
    **end for**
**end for**

---

## 4.2 Pre-training for non-dominated policies via iterative discovery

The analysis above inspires us to discover a refined policy class, $\widetilde{\Pi}$, during training. At test time, the relaxed definition, $\widetilde{\text{Regret}}(T)$, with respect to the refined policy class $\widetilde{\Pi}$ can be efficiently minimized. However, the gap between $\widetilde{\text{Regret}}(T)$ and $\text{Regret}(T)$ can be significant when policies in $\widetilde{\Pi}$ are suboptimal, meaning that policies from $\Pi \setminus \widetilde{\Pi}$ could provide much higher rewards against some attacks. Consequently, we introduce the following definition to characterize the optimality of $\widetilde{\Pi}$.

**Definition 4.6.** *For given policy class $\widetilde{\Pi}$, we define the optimality gap between $\widetilde{\Pi}$ and $\Pi$ as*

$$\text{Gap}(\widetilde{\Pi}, \Pi) := \max_{\nu \in \mathcal{V}} \left( \max_{\pi \in \Pi} J(\pi, \nu) - \max_{\pi' \in \widetilde{\Pi}} J(\pi', \nu) \right).$$

This definition implies that if we have $\text{Gap}(\widetilde{\Pi}, \Pi) \leq \epsilon$, then whatever policy the attacker uses, the optimal policy in $\widetilde{\Pi}$ is also $\epsilon$-optimal in $\Pi$. With this quantity, we can relate the two notions of regret.

**Proposition 4.7.** *Given $\widetilde{\Pi}$, it holds that for any $T \in \mathbb{N}$, $\{\pi^t\}_{t \in [T]}$, and $\{\nu^t\}_{t \in [T]}$*

$$\frac{\text{Regret}(T)}{T} \leq \frac{\widetilde{\text{Regret}}(T)}{T} + \text{Gap}(\widetilde{\Pi}, \Pi).$$

*If $|\widetilde{\Pi}| < \infty$, Algorithm 1 satisfies $\mathbb{E}[\text{Regret}(T)]/T \leq 2H\sqrt{\frac{|\widetilde{\Pi}| \log |\widetilde{\Pi}|}{T}} + \text{Gap}(\widetilde{\Pi}, \Pi)$.*

According to this proposition, there is a clear trade-off between optimality, i.e., $\text{Gap}(\widetilde{\Pi}, \Pi)$, and efficiency, i.e., $|\widetilde{\Pi}|$. A natural question arises: can we achieve a small $\text{Gap}(\widetilde{\Pi}, \Pi)$ while $\widetilde{\Pi}$ is finite? Indeed, we answer this in the affirmative.

**Proposition 4.8.** *There exists $\widetilde{\Pi}$ such that $\text{Gap}(\widetilde{\Pi}, \Pi) = 0$ while $|\widetilde{\Pi}| < \infty$.*

This confirms that we can always find an optimal $\widetilde{\Pi}$ with *finite* cardinality, enabling the execution of Algorithm 1. However, $|\widetilde{\Pi}|$ in our construction is contingent on the deterministic policy set, which is relatively large. This indeed arises because an optimal $\widetilde{\Pi}$ can also encompass many *redundant* policies. Removing these redundant policies from $\widetilde{\Pi}$ does not impact its optimality. To characterize such redundant policies, we define dominated policies as follows.

**Definition 4.9** (Dominated and Non-dominated Policy)**.** *Given $\delta \geq 0$ and $\widetilde{\Pi}$. We define $(\delta, \widetilde{\Pi})$-dominated policy $\pi \notin \widetilde{\Pi}$ as that there exists some $\omega \in \Delta(\widetilde{\Pi})$, for any $\nu \in \mathcal{V}$, $J(\pi, \nu) \leq \mathbb{E}_{\pi' \sim \omega}[J(\pi', \nu)] + \delta$. For $\delta = 0$, we also say $\pi$ is dominated by $\widetilde{\Pi}$. If $\pi$ is not a $(0, \widetilde{\Pi} \setminus \{\pi\})$-dominated policy, we say $\pi$ is a non-dominated policy (w.r.t $\widetilde{\Pi}$).*

It's clear that for a $(\delta, \widetilde{\Pi})$-dominated policy $\pi$, i.e., $\min_{\omega \in \Delta(\widetilde{\Pi})} \max_\nu \left( J(\pi, \nu) - \mathbb{E}_{\pi' \sim \omega}[J(\pi', \nu)] \right) \leq \delta$, including $\pi$ in $\widetilde{\Pi}$ allows the optimality gap to decrease by at most $\delta$. With this principle, a

straightforward algorithm to construct a small and optimal policy class is to start from an optimal $\widetilde{\Pi}$ (potentially with redundant policies), i.e., $\mathrm{Gap}(\widetilde{\Pi}, \Pi) = 0$, and then enumerate all $\pi \in \widetilde{\Pi}$ to examine whether $\pi$ is dominated by $\widetilde{\Pi} \setminus \pi$. If it is true, one can remove $\pi$ from $\widetilde{\Pi}$ to reduce its cardinality. This process is akin to (iterated) elimination of dominated actions in normal-form games (Roughgarden, 2010).

While such procedures can maintain the optimality of $\widetilde{\Pi}$ and effectively reduce its cardinality, the overhead of enumerating all $\pi \in \widetilde{\Pi}$ can be unacceptable. Consequently, a natural and more efficient approach is to construct $\widetilde{\Pi}$ *from scratch* by iteratively expanding $\widetilde{\Pi}$. Specifically, given $\widetilde{\Pi}$, any policy $\pi$ such that $\min_{\omega \in \Delta(\widetilde{\Pi})} \max_\nu \left( J(\pi, \nu) - \mathbb{E}_{\pi' \sim \omega}[J(\pi', \nu)] \right) > \delta$ can be used to expand $\widetilde{\Pi}$. Thus, we propose to select the one that *maximizes* this quantity $\min_{\omega \in \Delta(\widetilde{\Pi})} \max_\nu \left( J(\pi, \nu) - \mathbb{E}_{\pi' \sim \omega}[J(\pi', \nu)] \right)$. In other words, at each iteration $k$, given $\widetilde{\Pi}^k = \{\pi^1, \cdots, \pi^k\}$ already discovered, we solve the following optimization problem:

$$\pi^{k+1} \in \arg \max_{\pi \in \Pi} \min_{\omega \in \Delta(\{\pi^1, \cdots, \pi^k\})} \max_{\nu \in \mathcal{V}} \left( J(\pi, \nu) - \mathbb{E}_{\pi' \sim \omega}[J(\pi', \nu)] \right), \quad (4.2)$$

$$f_{k+1} = \max_{\pi \in \Pi} \min_{\omega \in \Delta(\{\pi^1, \cdots, \pi^k\})} \max_{\nu \in \mathcal{V}} \left( J(\pi, \nu) - \mathbb{E}_{\pi' \sim \omega}[J(\pi', \nu)] \right).$$

It turns out such an iterative process enjoys guarantees for both *optimality and efficiency*.

**Theorem 4.10.** *For any $\delta > 0$, there exists $K \in \mathbb{N}$ such that $f_K \leq \delta$. Correspondingly, the policy class $\widetilde{\Pi}^K := \{\pi^1, \cdots, \pi^K\}$ satisfies that $\mathrm{Gap}(\widetilde{\Pi}^K, \Pi) \leq \delta$. Furthermore, we have the regret guarantee that $\mathbb{E}[\mathrm{Regret}(T)]/T \leq 2H\sqrt{\frac{K \log K}{T}} + \delta$ for Algorithm 1.*

*Moreover, let $K^\star = \min_{\mathrm{Gap}(\widetilde{\Pi}, \Pi)=0} |\widetilde{\Pi}|$ and $K^{fin} = \min_{K \in \mathbb{N}: f_K = 0} K$, as long as our objective 4.2 admits a unique solution at every iteration, our algorithm finishes within at most $K^\star + 1$ iterations, i.e., we have $K^{fin} \leq K^\star + 1$.*

**Implications.** The first part of Theorem 4.10 implies that we can simply set an error threshold $\delta > 0$ and sequentially solve Equation 4.2 until the optimal value is less than or equal to $\delta$. Then, Theorem 4.10 predicts this process will always finish in finite iterations, thus leading to a finite $\widetilde{\Pi}$ for any given $\delta$. Once it converges, it is guaranteed that $\mathrm{Gap}(\widetilde{\Pi}, \Pi) \leq \delta$. In addition, the second part of Theorem 4.10 proves that, under mild conditions, once the algorithm discovers a $\widetilde{\Pi}$ such that the optimality gap is 0, $\widetilde{\Pi}$ is guaranteed to be the smallest one.

**A practical algorithm.** To solve the objective 4.2 and develop a practical algorithm, we leverage the fact by weak duality that

$$\max_{\pi \in \Pi} \min_{\omega \in \Delta(\{\pi^1, \cdots, \pi^k\})} \max_{\nu \in \mathcal{V}} \left( J(\pi, \nu) - \mathbb{E}_{\pi' \sim \omega}[J(\pi', \nu)] \right)$$
$$\geq \max_{\pi \in \Pi} \max_{\nu \in \mathcal{V}} \min_{\omega \in \Delta(\{\pi^1, \cdots, \pi^k\})} \left( J(\pi, \nu) - \mathbb{E}_{\pi' \sim \omega}[J(\pi', \nu)] \right).$$

Therefore, we propose to optimize RHS, a lower bound for the original problem, bringing two benefits: (1) the maximization for $\pi$ and $\nu$ can be merged and updated together (2) the inner minimization problem is tractable. To solve RHS, we follow the common practice for nonconcave-convex optimization problems, repeating the process of first solving the inner problem exactly, and then running gradient ascent for the outer max problem (Lin et al., 2020). The detailed algorithm is presented in Algorithm 2. **Notably, the attacker $\nu$ is not modeled as the worst-case to minimize the victim rewards anymore.** For a more intuitive illustration, we refer to the left part of Figure 1.

Finally, to deepen the understanding of our problem and algorithm, we provide a negative result regarding $|\widetilde{\Pi}|$. In Theorem 4.10, we have not shown how $K^{\mathrm{fin}}$ explicitly depends on $\delta$ or other problem parameters $(S, A, H)$. Indeed, this is not a caveat of our algorithm or analysis. We point out in the following theorem that, for some problems, $\widetilde{\Pi}$ must be large to be near-optimal.

**Theorem 4.11.** *There exists an MDP with $S = 2$, $A = 2$ such that for any $|\widetilde{\Pi}| < 2^H$, we must have $\mathrm{Gap}(\widetilde{\Pi}, \Pi) \geq \frac{1}{4}$.*

---

**Algorithm 2** Iterative discovery of non-dominated policy class

---

**Input:** $\delta, \eta_1, \eta_2, K, N$
Initialize $\widetilde{\Pi}^1 \leftarrow \{\pi^1\}$, $k \leftarrow 1$, $f_k \leftarrow \infty$
**for** $k = 1, \cdots, K$ iterations **do**
    Initialize $\pi^{k+1,0}$, $\nu^0$, $t \leftarrow 0$, and $f_{k+1} \leftarrow 0$
    **for** $t = 1, \cdots, N$ iterations **do**
        $k^\star \leftarrow \arg\max_{k' \in [k]} J(\pi^{k'}, \nu^t)$         ▷ estimating accumulative rewards with samples
        $\nu^{t+1} \leftarrow \nu^t + \eta_1 \nabla_\nu (J(\pi^{k+1,t}, \nu^t) - J(\pi^{k^\star}, \nu^t))$   ▷ updating with SA-RL (Zhang et al.,
2021) or PA-AD (Sun et al., 2021)
        $\pi^{k+1,t+1} \leftarrow \pi^{k+1,t} + \eta_2 \nabla_\pi J(\pi^{k+1,t}, \nu^t)$              ▷ updating with PPO
        $f_{k+1} \leftarrow J(\pi^{k+1,t+1}, \nu^{t+1}) - J(\pi^{k^\star}, \nu^{t+1})$
        $t \leftarrow t + 1$
    **end for**
    $\pi^{k+1} \leftarrow \pi^{k+1,t}$
    $\widetilde{\Pi}^{k+1} \leftarrow \widetilde{\Pi}^k \cup \{\pi^{k+1}\}$
**end for**

---

Nevertheless, this does not mean the problem is always intractable. As for concrete applications, it is possible that $f_k$ can still converge to a small value quickly as $k$ increases. Therefore, we shall investigate how the cardinality of $\widetilde{\Pi}$ affects empirical performance on standard benchmarks. We remark that Proposition 4.3 and Theorem 4.11 reveal the fundamental hardness of our problem setting for test time and training time respectively.

### 4.3 HOW TO ATTACK ADAPTIVE VICTIM POLICIES OPTIMALLY?

Although our primary focus is on developing robust victims against attacks beyond worst-case scenarios, we also explore how to attack an adaptive victim *optimally*. Existing works typically formulate this as a single-agent RL problem, as *the attacker usually targets only a single static victim in a stationary environment*. However, once the victim can adapt, the attack problem becomes more challenging. Since our focus is on developing robust victims, we consider a white-box attack setup, where the attacker is aware that the victim will be adaptive and will use the refined policy class $\widetilde{\Pi}$ at test time. Consequently, its attack objective can be framed as

$$\min_\nu \max_{\omega \in \Delta(\widetilde{\Pi})} \mathbb{E}_{\pi \sim \omega} J(\pi, \nu),$$

accounting for the fact that the victim can adaptively identify its optimal choice from $\widetilde{\Pi}$ in response to any arbitrary static attacker $\nu$, as per Proposition 4.5. While this objective might seem formidable to solve, it turns out that existing works have already laid the groundwork for this problem. In this context, the inner problem can be solved tractably, and the outer minimization problem can be addressed by employing existing RL-based methods, such as SA-RL (Zhang et al., 2021) and PA-AD (Sun et al., 2021). Consequently, we can repeat the process of solving the inner maximization first and then applying a gradient update for the outer minimization problem (Lin et al., 2019a).

## 5 EXPERIMENTS

In this section, our primary focus is to explore the following questions:

▷ Can our methods attain improved robustness against non-worst-case static attacks in comparison to formulations that explicitly optimize for worst-case performance, while maintaining comparable robustness against worst-case attacks?

▷ Can our methods render better test-time performance against dynamic attackers through online adaptation compared to baselines deploying a single, static victim?

▷ Are our methods capable of achieving competitive performance with a reasonably small $\widetilde{\Pi}$?

### 5.1 EXPERIMENTAL SETUP AND BASELINES

For empirical studies, we implement our framework in four Mujoco environments with continuous action spaces, specifically, Hopper, Walker2d, Halfcheetah, and Ant, adhering to a setup similar to

| Environment | Model | Natural Reward | Random | RS | SA-RL | PA-AD |
|---|---|---|---|---|---|---|
| **Hopper** state-dim: 11 $\epsilon$=0.075 | ATLA-PPO | $3291 \pm 600$ | $3165 \pm 576$ | $2244 \pm 618$ | $1772 \pm 802$ | $1232 \pm 350$ |
| | PA-ATLA-PPO | $3449 \pm 237$ | $3325 \pm 239$ | $3002 \pm 329$ | $1529 \pm 284$ | $2521 \pm 325$ |
| | WocaR-PPO | $3616 \pm 99$ | $3633 \pm 30$ | $3277 \pm 159$ | $2390 \pm 145$ | $2579 \pm 229$ |
| | **Ours** | $\mathbf{3652 \pm 108}$ | $\mathbf{3653 \pm 57}$ | $\mathbf{3332 \pm 713}$ | $\mathbf{2526 \pm 682}$ | $\mathbf{2896 \pm 723}$ |
| **Walker2d** state-dim: 17 $\epsilon$=0.05 | ATLA-PPO | $3842 \pm 475$ | $3927 \pm 368$ | $3239 \pm 294$ | $3663 \pm 707$ | $1224 \pm 770$ |
| | PA-ATLA-PPO | $4178 \pm 529$ | $4129 \pm 78$ | $3966 \pm 307$ | $3450 \pm 178$ | $2248 \pm 131$ |
| | WocaR-PPO | $4156 \pm 495$ | $4244 \pm 157$ | $4093 \pm 138$ | $3770 \pm 196$ | $2722 \pm 173$ |
| | **Ours** | $\mathbf{6319 \pm 31}$ | $\mathbf{6309 \pm 36}$ | $\mathbf{5916 \pm 790}$ | $\mathbf{6085 \pm 620}$ | $\mathbf{5803 \pm 857}$ |
| **Halfcheetah** state-dim: 17 $\epsilon$=0.15 | ATLA-PPO | $6157 \pm 852$ | $6164 \pm 603$ | $4806 \pm 392$ | $5058 \pm 418$ | $2576 \pm 548$ |
| | PA-ATLA-PPO | $6289 \pm 342$ | $6215 \pm 346$ | $5226 \pm 114$ | $4872 \pm 379$ | $3840 \pm 273$ |
| | WocaR-PPO | $6032 \pm 68$ | $5969 \pm 149$ | $5319 \pm 220$ | $\mathbf{5365 \pm 54}$ | $4269 \pm 172$ |
| | **Ours** | $\mathbf{7095 \pm 88}$ | $\mathbf{6297 \pm 471}$ | $\mathbf{5457 \pm 385}$ | $5089 \pm 86$ | $\mathbf{4411 \pm 718}$ |
| **Ant** state-dim: 111 $\epsilon$=0.15 | ATLA-PPO | $5359 \pm 153$ | $5366 \pm 104$ | $4136 \pm 149$ | $3765 \pm 101$ | $220 \pm 338$ |
| | PA-ATLA-PPO | $5469 \pm 106$ | $5496 \pm 158$ | $4124 \pm 291$ | $3694 \pm 188$ | $2986 \pm 364$ |
| | WocaR-PPO | $5596 \pm 225$ | $5558 \pm 241$ | $4339 \pm 160$ | $3822 \pm 185$ | $3164 \pm 163$ |
| | **Ours** | $\mathbf{5769 \pm 290}$ | $\mathbf{5630 \pm 146}$ | $\mathbf{4683 \pm 561}$ | $\mathbf{4524 \pm 79}$ | $\mathbf{4312 \pm 281}$ |

Table 1: Average episode rewards $\pm$ standard deviation over 50 episodes with three baselines on Hopper, Walker2d, Halfcheetah, and Ant. $\epsilon$ stands for the attack budget chosen to be the same as related works. We use $|\widetilde{\Pi}| = 5$ for ours and discuss its choice later. Natural reward and rewards under four types of attacks are reported. Under each column corresponding to an evaluation metric, we bold the best results. And the row for the most robust agent is highlighted as gray .

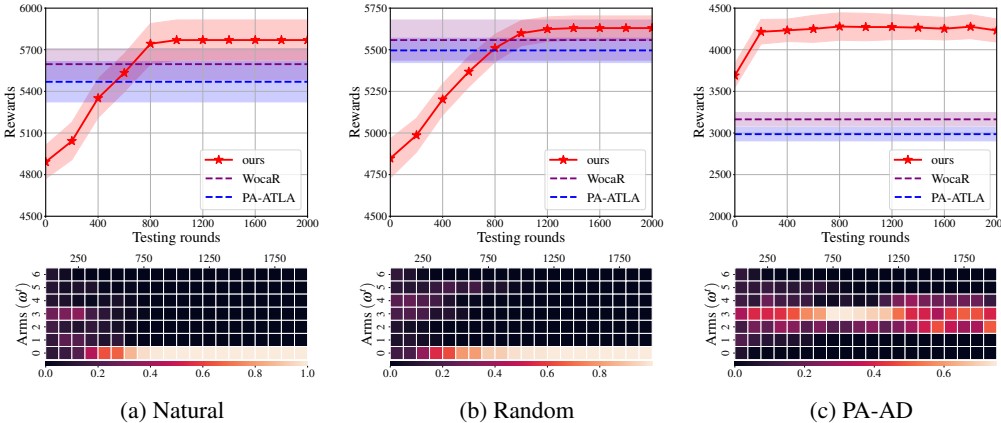

(a) Natural

(b) Random

(c) PA-AD

Figure 2: Online adaptation when facing unknown static attackers. It can be seen that the best policy can be identified quickly and reliably within 800 episodes or less against different attackers.

most related works (Zhang et al., 2020a; 2021; Sun et al., 2019; Liang et al., 2022). We compare our methods with several state-of-the-art robust training methods including ATLA-PPO (Zhang et al., 2021), PA-ATLA-PPO (Sun et al., 2021), and WocaR-PPO (Liang et al., 2022). WocaR-PPO is reported to be the most robust in most environments. We defer the comparison with other baselines, along with additional implementation and hyperparameter details to the Appendix.

## 5.2 PERFORMANCE AGAINST STATIC ATTACKS

In this subsection, we showcase improved performance against a spectrum of attacks, ranging from no attacks to the strongest ones. Accordingly, we present the natural rewards to depict the scenario without any attacks. We incorporate two heuristic attacks: random perturbations and robust SARSA (RS) (Zhang et al., 2020a), representing attacks beyond worst-case scenarios. We also include SA-RL attacks (Zhang et al., 2021) to reflect scenarios where the attacker might have limited algorithmic efficiency to devise an optimal attack policy since SA-RL can struggle with large action space in its formulation, and its attack performance can be further enhanced as indicated by Sun et al. (2021),

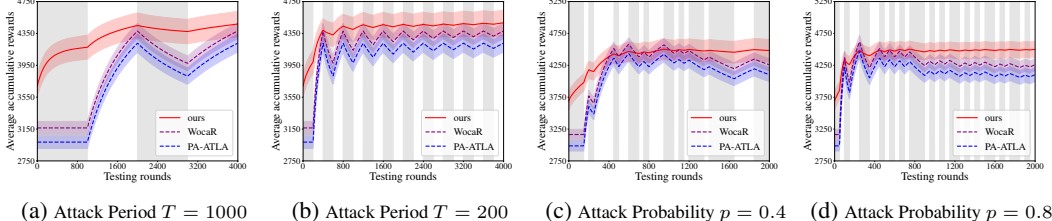

(a) Attack Period $T = 1000$     (b) Attack Period $T = 200$     (c) Attack Probability $p = 0.4$     (d) Attack Probability $p = 0.8$

Figure 3: Time averaged accumulative rewards during online adaptation against periodic and probabilistic switching attacks on Ant. The shaded area indicates PA-AD attacks are active while the unshaded area indicates no attacks.

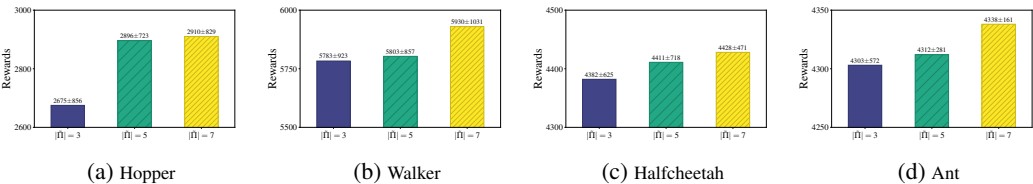

(a) Hopper      (b) Walker      (c) Halfcheetah      (d) Ant

Figure 4: Ablation study of $|\widetilde{\Pi}|$ against PA-AD attacks on $4$ environments.

making it non-worst-case as well. Lastly, we incorporate PA-AD, the currently strongest attack. **As observed in Table 1, our methods yield considerably higher natural rewards and consistently enhanced robustness against a spectrum of attacks.** To further show the improved performance against non-worst-case attacks, **we report the robustness under random attacks with various intensities in §E.3, where our methods are consistently better**. Given that our victim policy is adaptive, some additional adaptation steps might be necessary to identify the optimal policy against the attackers. To illustrate this, **we detail the adaptation process in Figure 2, showcasing that the best policy within $\widetilde{\Pi}$ can be identified rapidly and reliably.**

### 5.3   PERFORMANCE AGAINST DYNAMIC ATTACKS

We also examine scenarios where the attacker can exhibit dynamic behavior. To model such scenarios, we let attackers switch between no attacks and PA-AD attacks in the following two fashions.

**Periodic attacks.** Here we examine a mode where the attacker is weaker than in the worst-case scenarios, characterized by attacks appearing only periodically. We depict the performance against periodic attacks with varied frequencies.

**Probabilistic switching attacks.** In this section, we explore another mode where the attacker is less severe than in the worst-case scenarios. The attacker can toggle between being active and inactive. This switching is constrained to occur only with a probability $p$ at regular intervals.

The results are shown in Figure 3, illustrating that the average cumulative reward, or conversely, the negative of the regret, consistently outperforms the baselines.

### 5.4   ON THE SCALABILITY OF $|\widetilde{\Pi}|$

One major concern regarding our approach is that it may require a rather large policy class $\widetilde{\Pi}$ to achieve desirable performance. We report the performance of our methods with different $|\widetilde{\Pi}|$ against PA-AD attacks in Figure 4 on all environments. **Surprisingly, our methods only need within $3$ policies for $\widetilde{\Pi}$ to achieve improved performance compared with baselines.**

### 6   CONCLUDING REMARKS AND LIMITATIONS

In this paper, we develop a general framework to improve victim performance against attacks beyond worst-case scenarios. There are two phases: pre-training of non-dominated policies and online adaptation via no-regret learning. One limitation is the potentially high overhead during training (approximately $2\times$ running time compared with Sun et al. (2021); Liang et al. (2022)), as highlighted by Theorem 4.11. Additionally, identifying natural conditions to circumvent the hardness results outlined in Proposition 4.3 and Theorem 4.11, such as Lipschitz transition dynamics and rewards, is not fully addressed and remains an important topic for future works.

## 7 ACKNOWLEDGEMENT

Liu, Deng, Sun, Liang, and Huang are supported by National Science Foundation NSF-IISFAI program, DOD-ONR-Office of Naval Research, DOD Air Force Office of Scientific Research, DOD-DARPA-Defense Advanced Research Projects Agency Guaranteeing AI Robustness against Deception (GARD), Adobe, Capital One and JP Morgan faculty fellowships.

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

# Appendix for "Beyond Worst-case Attacks: Robust RL with Adaptive Defense via Non-dominated Policies"

# Table of Contents

## A  ADDITIONAL RELATED WORKS

**Other works related to adversarial RL.** Although our paper mainly studies the popular attack model of adversarial state perturbations, the vulnerability of RL is also studied under other different threat models. Adversarial action attacks are developed separately from state attacks including Pan et al. (2019); Tessler et al. (2019); Tan et al. (2020); Lee et al. (2021). Poisoning (Behzadan & Munir, 2017; Huang & Zhu, 2019; Sun et al., 2020; Zhang et al., 2020b; Rakhsha et al., 2020) is another type of adversarial attack that manipulates the training data, different from the test-time attacks that deprave a well-trained policy.

**Diverse multi-policy RL.** There are also a bunch of related works dedicated to developing RL policies that can generalize to unknown test environments. The main idea is to encourage the diversity of learned policies (Eysenbach et al., 2018; Kumar et al., 2020), by ensuring good coverage in the state occupancy space *for the training environment*. However, the robustness of such policies against malicious, and even adaptive attackers during test time remains an open question. We posit that incorporating the possibility of adaptive test-time attackers into the training phase is critical for developing robust policies. Meanwhile, Zahavy et al. (2021) considers constructing a diverse set of policies through a robustness objective, which targets the *worst-case reward*.

**Multi-objective RL and optimization.** In the training phase, the problem we investigate is conceptually similar to multi-objective RL, wherein the objective functions correspond to the victim's rewards against a range of potential attackers. Extant literature primarily adopts one of two approaches to this challenge (Roijers et al., 2013). The first approach converts the multi-objective problem into a single-objective optimization task through a variety of techniques, subsequently employing traditional algorithms to identify solutions (Kim & de Weck, 2006; Konak et al., 2006; Nakayama et al., 2009). However, such methods inherently yield an *average* policy over the preference space and lack the flexibility to optimize for individualized preference vectors. In contrast, our methodology during the training phase aligns more closely with the second category of approaches,

which seeks an optimal policy set that spans the entire domain of feasible preferences (Natarajan & Tadepalli, 2005; Barrett & Narayanan, 2008; Mossalam et al., 2016; Yang et al., 2019). Unfortunately, existing techniques are not well-suited to address the unique complexities of our problem. Specifically, conventional methods are predicated on the assumption that, in multi-objective RL, distinct objectives only alter the reward function of the MDP, while the transition dynamics remain invariant. This structure facilitates the use of established algorithms such as value iteration or Q-learning. In the context of our problem, as mentioned before, this assumption does not hold, as the attacker significantly influences the transition dynamics from the victim's standpoint.

# B   THEORETICAL ANALYSIS

## B.1   SUPPORTING LEMMAS

Here we prove the following series of lemmas for the proof of our propositions and theorems. From now on, for any $\omega \in \Delta(\Pi)$ and $\nu$, we use the shorthand notation $J(\omega, \nu) := \mathbb{E}_{\pi \sim \omega} J(\pi, \nu)$.

**Lemma B.1.** *For any $\pi \in \Pi$, there always exists $\omega \in \Delta(\Pi^{det})$ such that $J(\pi, \nu) = J(\omega, \nu)$ for any $\nu \in \mathcal{V}$.*

*Proof.* Consider any trajectory $\{s_h, \widehat{s}_h, a_h\}_{h \in [H]}$ and random seed $z \in \mathcal{Z}$, we compute its probability under policy $\pi \in \Pi$ and $\nu \in \mathcal{V}$ as follows

$$\mathbb{P}^{\pi, \nu}(\{s_h, \widehat{s}_h, a_h\}_{h \in [H]}, z)$$

$$= \mathbb{P}(z)\mu_1(s_1)\nu_1(\widehat{s}_1 \mid s_1, z)\pi(a_1 \mid \widehat{s}_1) \prod_{h=2}^{H} \mathbb{T}(s_h \mid s_{h-1}, a_{h-1})\nu_h(\widehat{s}_h \mid s_h, z)\pi(a_h \mid \widehat{s}_{1:h}, a_{1:h-1})$$

$$= \left[\pi(a_1 \mid \widehat{s}_1) \prod_{h=2}^{H} \pi(a_h \mid \widehat{s}_{1:h}, a_{1:h-1})\right] \mathbb{P}(z)\mu_1(s_1)\nu_1(\widehat{s}_1 \mid s_1, z) \prod_{h=2}^{H} \mathbb{T}(s_h \mid s_{h-1}, a_{h-1})\nu_h(\widehat{s}_h \mid s_h, z).$$

Now we are ready to construct the mixture of policy $\omega \in \Delta(\Pi^{\text{det}})$. For any $\pi' \in \Pi^{\text{det}}$, we define its probability in the mixture as

$$\omega(\pi') := \prod_{h' \in [H]} \prod_{\{\widehat{s}'_h, a'_h\}_{h \in [h']}} \pi(\pi'(\widehat{s}'_{1:h}, a'_{1:h-1}) \mid \widehat{s}'_{1:h}, a'_{1:h-1}). \tag{B.1}$$

Now we can compute

$$\mathbb{P}^{\omega, \nu}(\{s_h, \widehat{s}_h, a_h\}_{h \in [H]}, z) = \mathbb{E}_{\pi' \sim \omega} \mathbb{P}^{\pi', \nu}(\{s_h, \widehat{s}_h, a_h\}_{h \in [H]}, z)$$

$$= \left[\mathbb{P}(z)\mu_1(s_1)\nu_1(\widehat{s}_1 \mid s_1, z) \prod_{h=2}^{H} \mathbb{T}(s_h \mid s_{h-1}, a_{h-1})\nu_h(\widehat{s}_h \mid s_h, z)\right] \mathbb{E}_{\pi' \sim \omega} \mathbb{1}\left[a_1 = \pi'(\widehat{s}_1), \{a_h = \pi'(\widehat{s}_{1:h}, a_{1:h-1})\}_{h=2}^{H}\right]$$

$$= \left[\mathbb{P}(z)\mu_1(s_1)\nu_1(\widehat{s}_1 \mid s_1, z) \prod_{h=2}^{H} \mathbb{T}(s_h \mid s_{h-1}, a_{h-1})\nu_h(\widehat{s}_h \mid s_h, z)\right] \mathbb{P}(a_1 = \pi'(\widehat{s}_1), \{a_h = \pi'(\widehat{s}_{1:h}, a_{1:h-1})\}_{h=2}^{H})$$

$$= \left[\mathbb{P}(z)\mu_1(s_1)\nu(\widehat{s}_1 \mid s_1, z) \prod_{h=2}^{H} \mathbb{T}(s_h \mid s_{h-1}, a_{h-1})\nu_h(\widehat{s}_h \mid s_h, z)\right]\left[\pi(a_1 \mid \widehat{s}_1) \prod_{h=2}^{H} \pi(a_h \mid \widehat{s}_{1:h}, a_{1:h-1})\right],$$

where the last step comes from the construction of $\omega$ in Equation B.1 by marginalization. Therefore, we conclude that $\mathbb{P}^{\pi, \nu}(\{s_h, \widehat{s}_h, a_h\}_{h \in [H]}, z) = \mathbb{P}^{\omega, \nu}(\{s_h, \widehat{s}_h, a_h\}_{h \in [H]}, z)$, where construction of $\omega$ does not depend on $\nu$, proving our lemma. □

**Lemma B.2.** *The optimization problem of Equation 4.2 always admits a deterministic solution.*

*Proof.* Note by the definition of $\mathcal{V} := \Delta(\mathcal{V}^{\text{det}})$, indeed strong duality holds:

$$\max_{\pi^{k+1} \in \Pi} \min_{\omega \in \Delta(\{\pi^1, \cdots, \pi^k\})} \max_{\nu \in \mathcal{V}} \left(J(\pi^{k+1}, \nu) - \mathbb{E}_{\pi' \sim \omega}[J(\pi', \nu)]\right)$$

$$= \max_{\pi^{k+1} \in \Pi} \max_{\nu \in \mathcal{V}} \min_{\omega \in \Delta(\{\pi^1, \cdots, \pi^k\})} \left(J(\pi^{k+1}, \nu) - \mathbb{E}_{\pi' \sim \omega}[J(\pi', \nu)]\right).$$

Then for any $\pi^{k+1,\star}, \nu^\star \in \arg\max_{\pi^{k+1}\in\Pi, \nu\in\mathcal{V}} \min_{\omega\in\Delta(\{\pi^1,\cdots,\pi^k\})} \left(J(\pi^{k+1},\nu) - \mathbb{E}_{\pi'\sim\omega}[J(\pi',\nu)]\right)$, we denote $\pi^\star(\nu^\star) := \arg\max_{\pi^{k+1}\in\Pi} J(\pi^{k+1},\nu^\star)$. Note that $\pi^\star(\nu)$ can be always selected to be a deterministic policy by Lemma B.1. Meanwhile, it is easy to see that since $\pi^{k+1,\star}, \nu^\star$ is an optimal solution, $\pi^\star(\nu^\star), \nu^\star$ is also an optimal solution, i.e.,

$$\pi^\star(\nu^\star), \nu^\star \in \arg\max_{\pi^{k+1}\in\Pi, \nu\in\mathcal{V}} \min_{\omega\in\Delta(\{\pi^1,\cdots,\pi^k\})} \left(J(\pi^{k+1},\nu) - \mathbb{E}_{\pi'\sim\omega}[J(\pi',\nu)]\right),$$

concluding our lemma. $\qquad\square$

**Lemma B.3.** *Let $K \in \mathbb{N}$ be the integer such that $f_{K+1} = 0$ and $f_K > 0$. For any $2 \le k \le K$, there does not exist some $\omega^\star \in \Delta(\Pi^{det} \setminus \{\pi^k\})$ such that $\max_{\nu\in\mathcal{V}} \left(J(\pi^k,\nu) - J(\omega^\star,\nu)\right) \le 0$.*

*Proof.* To begin with, it is easy to see that there does not exist $1 \le k_1 < k_2 \le K$ such that $\pi^{k_1} = \pi^{k_2}$. This is because it will lead to the fact that $f_{k_2} = 0$. Now suppose there exists some $\omega^\star \in \Delta(\Pi^{det} \setminus \{\pi^k\})$ such that

$$\max_{\nu\in\mathcal{V}} \left(J(\pi^k,\nu) - J(\omega^\star,\nu)\right) \le 0.$$

This leads to the fact that

$$\min_{\omega\in\Delta(\{\pi^1,\cdots,\pi^{k-1}\})} \max_{\nu\in\mathcal{V}} \left(J(\pi^k,\nu) - J(\omega,\nu)\right) \le \min_{\omega\in\Delta(\{\pi^1,\cdots,\pi^{k-1}\})} \max_{\nu\in\mathcal{V}} \left(J(\omega^\star,\nu) - J(\omega,\nu)\right)$$

$$\le \max_{\omega'\in\Delta(\Pi^{det}\setminus\{\pi^k\})} \min_{\omega\in\Delta(\{\pi^1,\cdots,\pi^{k-1}\})} \max_{\nu\in\mathcal{V}} \left(J(\omega',\nu) - J(\omega,\nu)\right)$$

$$= \max_{\omega'\in\Delta(\Pi^{det}\setminus\{\pi^k\})} \max_{\nu\in\mathcal{V}} \min_{\omega\in\Delta(\{\pi^1,\cdots,\pi^{k-1}\})} \left(J(\omega',\nu) - J(\omega,\nu)\right)$$

$$= \max_{\pi\in\Pi^{det}\setminus\{\pi^k\}} \max_{\nu\in\mathcal{V}} \min_{\omega\in\Delta(\{\pi^1,\cdots,\pi^{k-1}\})} \left(J(\pi,\nu) - J(\omega,\nu)\right)$$

$$= \max_{\pi\in\Pi^{det}\setminus\{\pi^k\}} \min_{\omega\in\Delta(\{\pi^1,\cdots,\pi^{k-1}\})} \max_{\nu\in\mathcal{V}} \left(J(\pi,\nu) - J(\omega,\nu)\right),$$

where the second last step comes from exactly the same as the proof of Lemma B.2. This contradicts the fact that $\pi^k$ is the unique optimal solution at iteration $k$. $\qquad\square$

### B.2 PROOF OF PROPOSITION 4.3

*Proof.* We construct the MDP $\mathcal{M}$ with the state space $\mathcal{S} = \{s^{good}, s^{bad}, s^{dummy}\}$, action space $\mathcal{A} = \{a^{good}, a^{bad}\}$. For the reward, we define $r_h(\cdot,\cdot) = 0$ for $h \in [H-1]$ and $r_H(s^{good},\cdot) = 1$ and $r_H(s^{bad},\cdot) = 0$. For the transition, we define $\mathbb{T}(s^{good} \,|\, s^{good}, a^{good}) = 1$, $\mathbb{T}(s^{bad} \,|\, s^{good}, a^{bad}) = 1$, $\mathbb{T}(s^{bad} \,|\, s^{bad}, \cdot) = 1$. The initial state is always $s^{good}$. We consider the attacker's policy $\nu$ such that $\nu(s^{dummy} \,|\, \cdot) = 1$, which means the attacker deterministically perturbs the state to $s^{dummy}$. Therefore, for the victim to learn the optimal policy against such an attacker, it is equivalent to a multi-arm bandit problem with $2^H$ arms, for which the sample complexity of finding an approximately optimal policy must suffer from $\Omega(2^H)$. Meanwhile, if such a desirable regret in the proposition is possible, it means we can learn an $\epsilon$-optimal policy in such kind of multi-arm bandit problem with sample complexity $\text{poly}(S, A, H, \frac{1}{\epsilon})$, leading to the contradiction. $\qquad\square$

### B.3 PROOF OF PROPOSITION 4.7

*Proof.* For any $\nu^{1:T}$, we denote $\pi^\star \in \arg\max_{\pi\in\Pi} \frac{1}{T}\sum_{t=1}^T J(\pi,\nu^t)$. Then according to Definition 4.2, we have

$$\text{Regret}(T) = \sum_{t=1}^T \left(J(\pi^\star,\nu^t) - J(\pi^t,\nu^t)\right)$$

$$= \left(\sum_{t=1}^T J(\pi^\star,\nu^t) - \max_{\pi\in\widetilde{\Pi}}\sum_{t=1}^T J(\pi,\nu^t)\right) + \max_{\pi\in\widetilde{\Pi}}\sum_{t=1}^T \left(J(\pi,\nu^t) - J(\pi^t,\nu^t)\right)$$

$$\le T\,\text{Gap}(\widetilde{\Pi},\Pi) + \widetilde{\text{Regret}}(T),$$

where the last step comes from choosing $\nu = \text{Unif}(\nu^{1:T})$ in Definition 4.6. $\qquad\square$

### B.4 PROOF OF PROPOSITION 4.8

*Proof.* Note since in this proposition, we only care about the existence of a finite $\widetilde{\Pi}$, we do not care about its efficiency, i.e., how large the constructed $\widetilde{\Pi}$ is. Indeed, we can consider $\Pi^{\text{det}}$, which is a finite policy class with cardinality $|\Pi^{\text{det}}| = \mathcal{O}((SA)^H)$. Now we verify the optimality of $\Pi^{\text{det}}$. For any $\nu \in \mathcal{V}$, assume $\pi^\star \in \arg\max_{\pi \in \Pi} J(\pi, \nu)$. Then by Lemma B.1, we have there exists an $\omega^\star \in \Delta(\Pi^{\text{det}})$ such that $J(\pi^\star, \nu) = \mathbb{E}_{\pi^{\text{det}} \sim \omega^\star} J(\pi^{\text{det}}, \nu)$. Now we choose $\pi^{\text{det},\star} = \arg\max_{\pi^{\text{det}} \in \omega^\star} J(\pi^{\text{det}}, \nu)$. Then we have $J(\pi^{\text{det},\star}, \nu) \geq \mathbb{E}_{\pi^{\text{det}} \sim \omega^\star} J(\pi^{\text{det}}, \nu) = J(\pi^\star, \nu)$. Therefore, we conclude that for any $\nu \in \mathcal{V}$, we have $\max_{\pi \in \Pi} J(\pi, \nu) = \max_{\pi \in \Pi^{\text{det}}} J(\pi, \nu)$. Therefore, $\text{Gap}(\Pi^{\text{det}}, \Pi) = 0$.

$\square$

### B.5 PROOF OF THEOREM 4.10

*Proof.* We begin with the proof for the part of the theorem. For $\delta > 0$ and any $i_1, i_2, \cdots, i_{|\mathcal{V}^{\text{det}}|} \in [[\lceil \frac{H}{\delta} \rceil]]$, we define the set $\mathcal{D}(i_1, \cdots, i_{|\mathcal{V}^{\text{det}}|}) = \{\pi \in \Pi \mid (i_j - 1)\delta \leq J(\pi, \nu_j) < i_j \delta, \forall j \in [|\mathcal{V}^{\text{det}}|]\}$. Then according to Pigeonhole principle, there must exist $K \in \mathbb{N}$ and $k \in [K]$ such that $\pi^{K+1} \in \mathcal{D}(i'_1, \cdots, i'_{|\mathcal{V}^{\text{det}}|})$ and $\pi^k \in \mathcal{D}(i'_1, \cdots, i'_{|\mathcal{V}^{\text{det}}|})$ for some $i'_1, i'_2, \cdots, i'_{|\mathcal{V}^{\text{det}}|} \in [[\lceil \frac{H}{\delta} \rceil]]$. Therefore, we conclude that $|J(\pi^{K+1}, \nu) - J(\pi^k, \nu)| \leq \delta$ for any $\nu \in \mathcal{V}^{\text{det}}$, and correspondingly for any $\nu \in \mathcal{V}$. This lead to that $f_{K+1} \leq \delta$. Now we are ready to show that $\text{Gap}(\widetilde{\Pi}^{K+1}, \Pi) \leq \delta$. For any $\nu \in \mathcal{V}$, we define $\pi^\star \in \arg\max_{\pi \in \Pi} J(\pi, \nu)$. Meanwhile, there exists $\omega \in \Delta(\widetilde{\Pi}^{K+1})$ such that $J(\pi^\star, \nu) \leq J(\omega, \nu) + \delta$ since $f_{K+1} \leq \delta$. This implies that $J(\pi^\star, \nu) - \max_{\pi' \in \widetilde{\Pi}^{K+1}} J(\pi', \nu) \leq \delta$, proving $\text{Gap}(\widetilde{\Pi}^{K+1}, \Pi) \leq \delta$.

Now we prove the second part of our theorem. Suppose $K^\star < K^{\text{fin}} - 1$, we denote the corresponding optimal policy set as $\Pi^\star = \{\widehat{\pi}^1, \cdots, \widehat{\pi}^{K^\star}\}$. By Lemma B.1, for any $k \in [K^\star]$, there exists a $\omega^k \in \Delta(\Pi^{\text{det}})$ such that

$$J(\widehat{\pi}^k, \nu) = \sum_{j=1}^{|\Pi^{\text{det}}|} \omega^k(\pi^j) J(\pi^j, \nu),$$

for any $\nu \in \mathcal{V}$, where we have abused our notation for $\{\pi^2, \cdots, \pi^{K^{\text{fin}}}\}$ to denote deterministic policies, which are policies discovered by our algorithm since according to Lemma B.2, those policies are different and deterministic. Now since $K^\star < K^{\text{fin}} - 1$, there exists some $2 \leq j \leq K^{\text{fin}}$ such that $\omega^k(\pi^j) \leq \frac{2}{3}$ for any $k \in [K^\star]$. Now we denote $\epsilon = \min_{\omega \in \Delta(\Pi^{\text{det}} \setminus \{\pi^j\})} \max_{\nu \in \mathcal{V}} (J(\pi^j, \nu) - J(\omega, \nu)) > 0$ by Lemma B.3, and let $\nu^\star \in \arg\max_{\nu \in \mathcal{V}} \min_{\omega \in \Delta(\Pi^{\text{det}} \setminus \{\pi^j\})} (J(\pi^j, \nu) - J(\omega, \nu))$. Therefore, it holds that $J(\pi^j, \nu^\star) \geq J(\pi, \nu^\star) + \epsilon$ for any $\pi \in \Delta(\Pi^{\text{det}} \setminus \{\pi^j\})$. Then we are ready to examine $\text{Gap}(\Pi^\star, \Pi)$ as follows:

$$\text{Gap}(\Pi^\star, \Pi) \geq \max_{\pi \in \Pi} J(\pi, \nu^\star) - \max_{\pi' \in \Pi^\star} J(\pi', \nu^\star) \geq J(\pi^j, \nu^\star) - \max_{\pi' \in \Pi^\star} J(\pi', \nu^\star) \geq \frac{\epsilon}{3} > 0,$$

contradicting that $\text{Gap}(\Pi^\star, \Pi) = 0$. $\square$

### B.6 PROOF OF THEOREM 4.11

*Proof.* Let us firstly consider a one-step MDP with state space $\mathcal{S} = \{s_1, s_2\}$, action space $\mathcal{A} = \{a_1, a_2\}$, reward function $r(s_1, a_1) = r(s_2, a_2) = 1$ otherwise 0, and $\mu_1(s_1) = \mu_1(s_2) = \frac{1}{2}$. Now assume the attacker can only choose two policies $\nu^{good}$ such that $\nu^{good}(s_1) = s_1, \nu^{good}(s_2) = s_2$, and $\nu^{bad}$ such that $\nu^{bad}(s_1) = s_2, \nu^{bad}(s_2) = s_1$. Let us consider four *basis* victim policies $\{\pi^1, \cdots, \pi^4\}$, which select the action $(a_1, a_2), (a_1, a_1), (a_2, a_1), (a_2, a_2)$ respectively for states $s_1$ and $s_2$. Then it holds that for any policy $\pi \in \Pi$, there exists $\alpha^j \in [0, 1]$ and $\sum_j \alpha^j = 1$ such that $J(\pi, \cdot) = \sum_{j=1}^4 \alpha^j J(\pi^j, \cdot)$ by Lemma B.1. Now we have either $\alpha^1 \leq \frac{1}{2}$ or $\alpha^3 \leq \frac{1}{2}$. Let us say $\alpha^1 \leq \frac{1}{2}$ and the case for $\alpha^3 \leq \frac{1}{2}$ can be proved similarly. Consider the case where the attacker takes the policy $\nu^{good}$. Then we have $J(\pi^1, \nu^{good}) - J(\pi, \nu^{good}) \geq 1 - (\frac{1}{2} + \frac{1}{2} \times \frac{1}{2}) = \frac{1}{4}$. Therefore, we conclude that if $|\widetilde{\Pi}| < 2$, we must have $\text{Gap}(\widetilde{\Pi}, \Pi) \geq \frac{1}{4}$.

Now let us extend it to the MDP with $H$ steps, where in the previous MDP, at each time step, the current state transits to the next two states with uniform probability regardless of the action taken. We consider the attacker's policies, where at each time step it uses the policy $\nu^{good}$ or $\nu^{bad}$, resulting in totally $2^H$ policies, $\{\nu^1, \cdots, \nu^{2^H}\}$. Similarly, we can define basis policies, which at each time step selects the policy from $\{\pi^1, \cdots, \pi^4\}$, ignoring the history information except the current observation (perturbed state). This results in a total of $4^H$ policies, for which we denote $\{\bar{\pi}^1, \cdots, \bar{\pi}^{4^H}\}$. Due to the transition dynamics we have defined, for any $\pi \in \Pi$, there exists some $\alpha^j(\pi) \in [0,1]$ and $\sum_j \alpha^j(\pi) = 1$ such that $J(\pi, \cdot) = \sum_{j=1}^{4^H} \alpha^j(\pi) J(\bar{\pi}^j, \cdot)$. W.L.O.G, we say policies $\bar{\pi}^{1:2^H}$ as all the policies only selecting policies from $\{\pi^1, \pi^3\}$ at each time step. Now consider any $\widetilde{\Pi} = \{\widetilde{\pi}^1, \widetilde{\pi}^2, \cdots, \widetilde{\pi}^K\}$ with $K < 2^H$. Then there must be some $m \in [2^H]$ such that $\alpha^m(\widetilde{\pi}^k) \le \frac{1}{2}$ for any $k \in [K]$. Let us say $\bar{\pi}^m$ is the policy always choosing $\pi^1$ at all time steps and correspondingly denote $\nu^\star$ as the policy always choosing $\nu^{good}$ at each step. Therefore, we have $J(\bar{\pi}^m, \nu^\star) - J(\widetilde{\pi}^k, \nu^\star) \ge H - (H - 1 + \frac{1}{2} + \frac{1}{2} \times \frac{1}{2}) = \frac{1}{4}$ for any $k \in [K]$. This concludes that $\mathrm{Gap}(\widetilde{\Pi}, \Pi) \ge \frac{1}{4}$. $\qquad\square$

## C  EXAMPLE AND DETAILED EXPLANATIONS OF ITERATIVE DISCOVERY

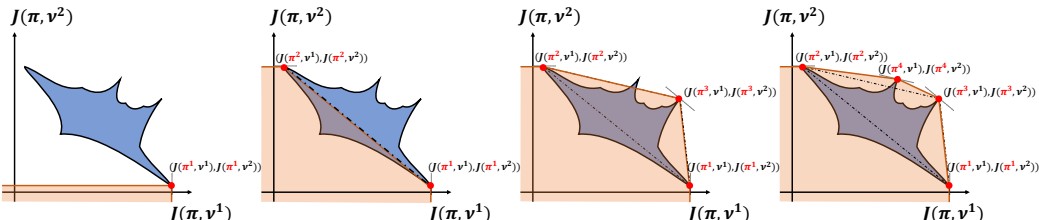

Figure 5: Iteration discovery of non-dominated policies in two dimensions.

Here we explain how our algorithm discovers the four policies $\pi^{1:4}$ in Figure 5, i.e., the left part of Figure 1. For simplicity, we consider there are only two pure attackers $\nu^1$ and $\nu^2$, and thus $\mathcal{V} = \Delta(\{\nu^1, \nu^2\})$.

**For the first iteration,** since there are no policies already discovered, the optimization problem we need to solve is $\pi^1 \in \arg\max_{\pi \in \Pi} \max_{\nu \in \mathcal{V}} J(\pi, \nu) = \arg\max_{\pi \in \Pi} \max\{J(\pi, \nu^1), J(\pi, \nu^2)\}$. By comparing $\nu^1$ and $\nu^2$, we can see the discovered policy is the rightmost one in Figure 5.

**For the second iteration,** given $\widetilde{\Pi} = \{\pi^1\}$ already discovered, the optimization problem we need to solve is $\pi^2 \in \arg\max_{\pi \in \Pi} \max_{\nu \in \mathcal{V}} \left(J(\pi, \nu) - J(\pi^1, \nu)\right)$. Since $\pi^1 \in \arg\max_{\pi \in \Pi} J(\pi, \nu^1)$, we have $\pi^2 \in \arg\max_{\pi \in \Pi} \max_{\nu \in \mathcal{V}} \left(J(\pi, \nu) - J(\pi^1, \nu)\right) = \arg\max_{\pi \in \Pi} \left(J(\pi, \nu^2) - J(\pi^1, \nu^2)\right) = \arg\max_{\pi \in \Pi} J(\pi, \nu^2)$. Therefore, $\pi^2$ is the uppermost one in Figure 5.

**For the third iteration,** given $\widetilde{\Pi} = \{\pi^1, \pi^2\}$ already discovered, the optimization problem we need to solve is $\pi^3 \in \arg\max_{\pi \in \Pi} \min_{\omega \in \Delta(\{\pi^1, \pi^2\})} \max_{\nu \in \mathcal{V}} \left(J(\pi, \nu) - J(\pi^1, \nu)\right)$. It is easy to see that in Figure 5, the optimal solution should be the one that's farthest from the line segment between $\pi^1$ and $\pi^2$. To see the reason, we can find that the optimal $\omega$ will be the point on the line segment between $\pi^1$ and $\pi^2$ such that $J(\pi^3, \nu^1) - J(\omega, \nu^1) = (\pi^3, \nu^2) - J(\omega, \nu^2)$.

**For the fourth iteration,** given $\widetilde{\Pi} = \{\pi^1, \pi^2, \pi^3\}$ already discovered, the optimization problem we need to solve is $\pi^4 \in \arg\max_{\pi \in \Pi} \min_{\omega \in \Delta(\{\pi^1, \pi^2, \pi^3\})} \max_{\nu \in \mathcal{V}} \left(J(\pi, \nu) - J(\pi^1, \nu)\right)$. From Figure 5, the optimization for $\omega$ will not put mass on policy $\pi^1$. Thus, what we need to solve is $\pi^4 \in \arg\max_{\pi \in \Pi} \min_{\omega \in \Delta(\{\pi^2, \pi^3\})} \max_{\nu \in \mathcal{V}} \left(J(\pi, \nu) - J(\pi^1, \nu)\right)$. Under the same reason as the third iteration, $\pi^4$ will be the one that is farthest to the line segment between $\pi^2$ and $\pi^3$.

Finally, it is worth mentioning that the analysis above holds only specifically (and roughly) for the reward landscape of Figure 5, for which we have simplified significantly to convey the intuitions. Actual problems we aim to deal with can be much more complex.

## D    DETAILS OF EXPERIMENTAL SETTINGS

In this section, we provide details of implementation and training hyperparameters for MuJoCo experiments. All experiments are conducted on NVIDIA GeForce RTX 2080 Ti GPU.

**Implementation details.** For the network structure, we employ a single-layer LSTM with 64 hidden neurons in Ant and Halfcheetah, and the original fully connected MLP structure in the other two environments. Both the victims and the attackers are trained with independent value and policy optimizers by PPO.

**Victim training.** For the baseline methods, we directly utilize the well-trained models for ATLA-PPO (Zhang et al., 2021), PA-ATLA-PPO (Sun et al., 2021), and WocaR-PPO (Liang et al., 2022) provided by the authors.

For the iterative discovery in Algorithm 2, we employ PA-AD to update attack models $\nu^t$ and PPO to update the victim. For the first policy $\pi^1$ in $\widetilde{\Pi}$, we train for 5 million steps (2441 iterations) in Ant and 2.5 million steps (1220 iterations) in the other three environments. For subsequent policies, we use the previously trained policy as the initialization and train for half of the steps of the first iteration to accelerate training.

Due to the high variance in RL training, the reported results are reported as the median performance from 21 agents trained with the same set of hyperparameters for reproducibility.

**Attack training.** The reported results under RS attack are from 30 trained robust value functions.

For evasion attacks such as SA-RL and PA-AD, we conduct a grid search of the optimal hyperparameters (including learning rates for the policy network and the adversary policy network, the ratio clip for PPO, and the entropy regularization) for each victim training method. We train for 10 million steps (4882 iterations) in Ant and 5 million steps (2441 iterations) in the other three environments. The reported results are from the strongest attack among all 108 trained adversaries.

## E    ADDITIONAL EXPERIMENTAL RESULTS

### E.1    ROBUSTNESS AGAINST VARIOUS DYNAMIC ATTACKS

In this section, we present the supplementary results demonstrating the robustness of our methods against various dynamic attacks. Two modes of dynamic attacks, periodic attacks, and probabilistic switching attacks, have been briefly introduced in §5.3. Here we show more details and results corresponding to these two dynamic attack modes.

**Periodic attacks.** We adjust the attack period $T$ from 1000 to 100 and examine the performance of our methods alongside two baselines. Additionally, we use a non-fixed period where $T$ alternates between 500 and 1000.

The average accumulative rewards and evolution of policy weights $\omega^t$ are shown in plots and heat maps in §6. Our observations are as follows: **(1)** Regardless of the duration of the periods, our methods consistently achieve higher average accumulative rewards than the two baseline methods. This underscores the efficacy of online adaptation in Algorithm 1. **(2)** The values of $\omega^t$ exhibit noticeable shifts during each period, highlighting the online adaptation process. **(3)** Even when $T$ alternates, our methods maintain their superiority over the baselines. The evolution of $\omega^t$ shows that our methods can effectively perceive the transition between two periods.

**Probabilistic attack.** We adjust the switching probability $p$ from 0.2 to 0.8. A higher value of $p$ signifies more frequent switching. We anticipate that it will be more challenging for the online adaptation of the agent. We keep the interval between two potential switching points as 50 rounds.

The results are exhibited in Figure 7, showcasing both the average accumulative rewards and the evolution of the weight $\omega^t$. We conclude that: **(1)** Our methods consistently outpace the two baselines. The superiority becomes more pronounced as the value of $p$ increases. **(2)** In contrast to the scenario with periodic attacks, the weights $\omega^t$ display a more random evolution. Nonetheless, they effectively converge to the arms yielding higher rewards.

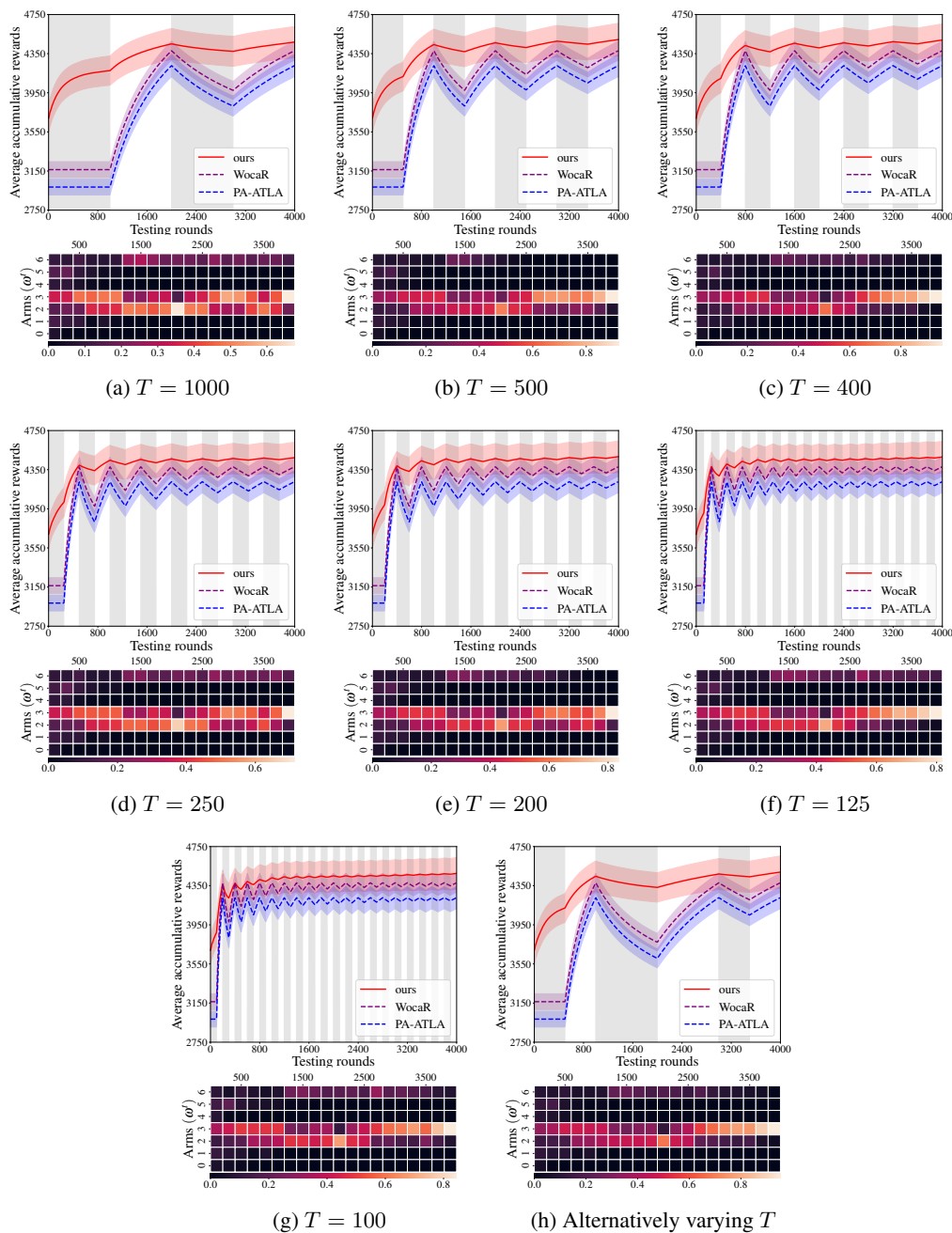

Figure 6: Time averaged accumulative rewards during online adaptation against periodic attacks on Ant. The shaded area showed in the indicates PA-AD attacks are active while the unshaded area indicates no attacks. The evolution of corresponding weights $\omega^t$ is shown in the heatmap where the brighter color means the higher value.

## E.2 ABLATION STUDY ON THE SCALABILITY OF $|\widetilde{\Pi}|$

A potential concern for our methods is the high computational cost of iterative discovery, which could render them impractical. To tackle this concern, we assess our methods using different scales of the policy class $|\widetilde{\Pi}|$ under PA-AD attacks across all four environments. The original value of $|\widetilde{\Pi}|$ in Table 1 is set to 5, and we modify it to both 3 and 7 for this ablation study. All other experimental parameters remain the same.

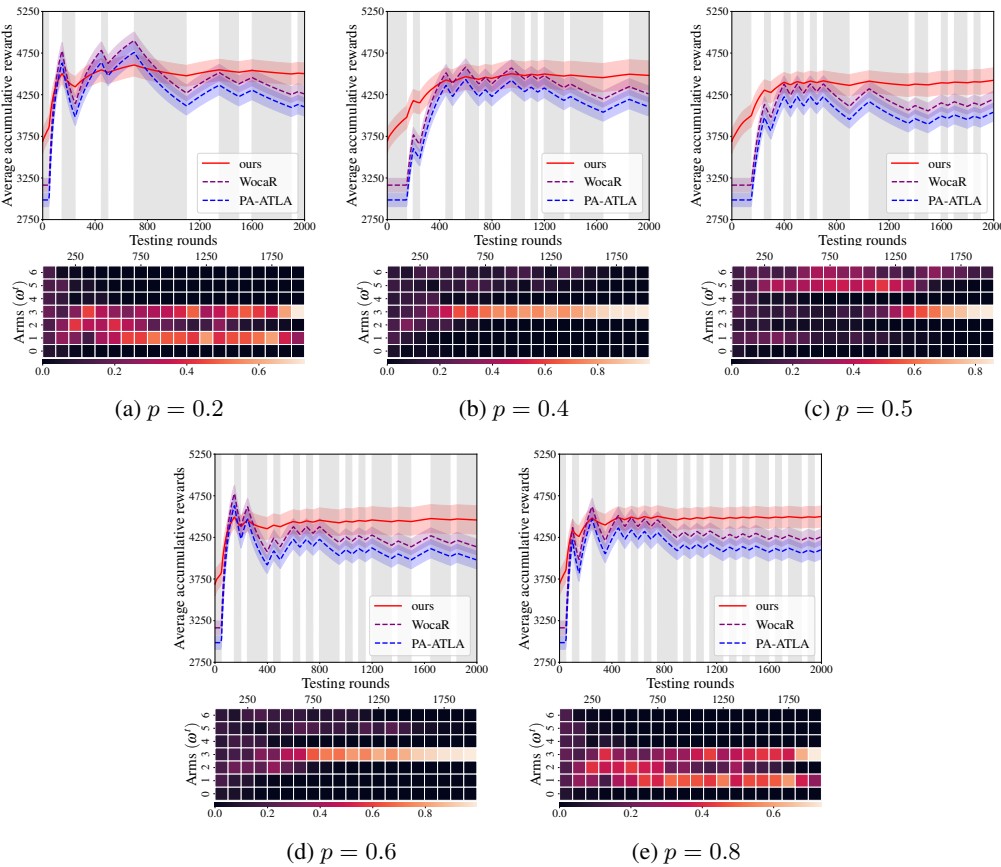

Figure 7: Time averaged accumulative rewards during online adaptation against probabilistic switching attacks on Ant. The shaded area showed in the indicates PA-AD attacks are active while the unshaded area indicates no attacks. The evolution of corresponding weights $\omega^t$ is shown in the heatmap where the brighter color means the higher value.

The results are depicted in Figure 8. We notice that: **(1)** The larger scale leads to higher rewards in all four environments. This implies that the non-dominated policy class, as it expands via iterative discovery, approaches the optimal one more accurately with increasing scales. **(2)** Even with a relatively modest scale of 3, our methods outpace the baseline methods in Table 1. This alleviates concerns about our new methods being reliant on unaffordable computational costs.

### E.3 ABLATION STUDY ON THE ATTACK BUDGET $\epsilon$

To examine how our methods perform under attacks with different values of the attack budget $\epsilon$, we evaluate their performance under a random attack across all four environments and compare them with two baselines. From Table 1, we observe that the random attack is relatively mild. However, its impact can be much worse if the attack budget is higher. Our goal is to evaluate the robustness of against non-worst-case attacks across various spectra.

The corresponding results are displayed in Figure 9. We derive the following observations: **(1)** When $\epsilon$ is small, the rewards of our methods are slightly higher than the baseline methods in nearly all environments. The exception is on Walker2d, where our methods distinctly outperform the baselines. It indicates the effectiveness of our methods in relatively clean environments. **(2)** As $\epsilon$ becomes moderate and continues to increase, although the performances of our methods decrease as PA-ATLA and WocaR, the rate of decline is slower compared to the two baseline methods. Previously, we only consider the non-worst-case attacks with the same $\epsilon$ by different modes. In this context, increasing values of $\epsilon$ for the same attack can be also interpreted as another non-worst-case attack. Thus, the

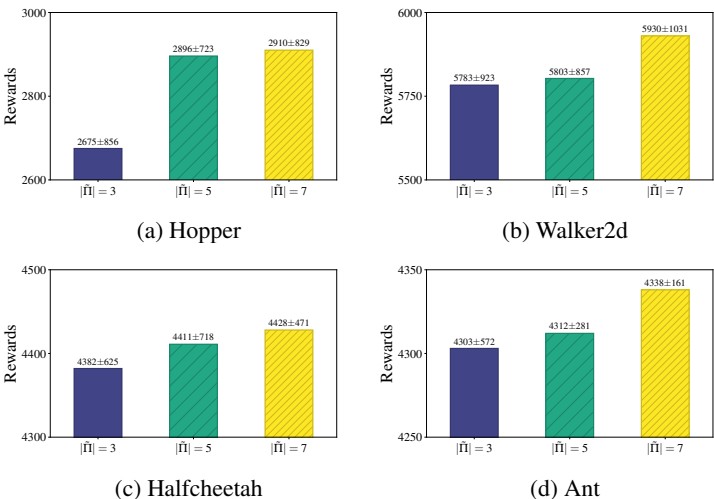

Figure 8: The performance for our methods with different non-dominant policy class scales $|\widetilde{\Pi}|$ in all four environments.

high rewards of our methods confirm their enhanced robustness against various types of non-worst-case attacks. **(3)** When $\epsilon$ is large, our methods continue to hold an advantage over the baseline methods. The only exception is Hopper, where the rewards from all three methods are nearly identical. This suggests that our new methods compromise little in terms of robustness against worst-case attacks.

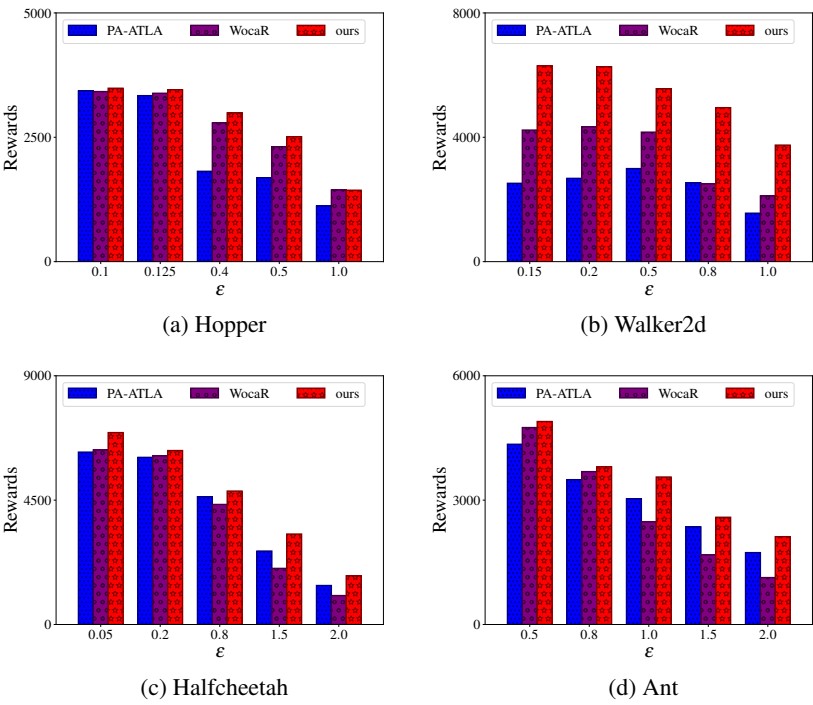

Figure 9: The performance for our methods and two baseline methods under attacks with different $\epsilon$ in all four environments.

### E.4 ABLATION STUDY ON THE WORST-CASE ROBUSTNESS

Although our primary goal is to improve the robustness against attacks beyond the worst cases, surprisingly, we find the robustness of our approach against currently strongest attacks PA-AD is also improved. To understand such reasons, we firstly notice that although related works including Zhang et al. (2021); Sun et al. (2021); Liang et al. (2022) share the same objective of explicitly maximizing the worst-case performance, Sun et al. (2021) improves over Zhang et al. (2021) and Liang et al. (2022) improves over Sun et al. (2021). **Therefore, we make the hypothesis that baseline approach may not have found the global optimal solution for their objective of maximizing robustness against the worst-case attacks reliably.** Specifically, one possible explanation from the perspective of optimization is that baselines could often converge to a local optima, while our approach explicitly encourages the new policy to *behave differently in the reward space compared with policies discovered already*, thus not stuck at a single local optimal solution easily. To further verify our hypothesis, we design the experiments as follows.

Firstly, note the number we report in Table 1 is a median number of 21 agents trained with the same set of hyperparameters following the set up of Zhang et al. (2021); Sun et al. (2021); Liang et al. (2022). To verify our hypothesis, we compare the performance of the best one and the median one of different approaches in Table 2. We can see that baselines like Sun et al. (2021); Liang et al. (2022) can match the high performance of ours in terms of the best run, while the median is low by a large margin. This means it is possible for baselines to achieve high worst-case robustness occasionally as its objective explicitly encourages so, but not reliably. In contrast, our methods are much more stable. This effectively supports our hypothesis.

| Environment | Model | PA-AD (Median) | PA-AD (Highest) |
|---|---|---|---|
| **Hopper** state-dim: 11 $\epsilon$=0.075 | PA-ATLA-PPO | $2521 \pm 325$ | $3129 \pm 316$ |
| | WocaR-PPO | $2579 \pm 229$ | $3284 \pm 193$ |
| | **Ours** | $2896 \pm 723$ | $3057 \pm 809$ |
| **Walker2d** state-dim: 17 $\epsilon$=0.05 | PA-ATLA-PPO | $2248 \pm 131$ | $3561 \pm 357$ |
| | WocaR-PPO | $2722 \pm 173$ | $4239 \pm 295$ |
| | **Ours** | $4239 \pm 295$ | $6052 \pm 944$ |
| **Halfcheetah** state-dim: 17 $\epsilon$=0.15 | PA-ATLA-PPO | $3840 \pm 273$ | $4260 \pm 193$ |
| | WocaR-PPO | $4269 \pm 172$ | $4579 \pm 368$ |
| | **Ours** | $4411 \pm 718$ | $4533 \pm 692$ |
| **Ant** state-dim: 111 $\epsilon$=0.15 | PA-ATLA-PPO | $2986 \pm 364$ | $3529 \pm 782$ |
| | WocaR-PPO | $3164 \pm 163$ | $4273 \pm 530$ |
| | **Ours** | $4273 \pm 530$ | $4406 \pm 329$ |

Table 2: Average episode rewards $\pm$ standard deviation with two baselines on Hopper, Walker2d, Halfcheetah, and Ant. The median and highest performance from 21 agents trained with the same set of hyperparameters are reported in two columns respectively.

### E.5 ABLATION STUDY ON ONLINE REWARD FOR BASELINES.

To clarify our novel Algorithm 2 to minimize $\mathrm{Gap}(\widetilde{\Pi}, \Pi)$ contributes to the major improvements of our approach instead of simply using online reward feedback, we conduct further ablation studies by also allowing baselines to use online reward feedbacks through our Algorithm 1. However, all baselines are essentially a single victim policy instead of a set, making it trivial to run Algorithm 1 since it will only have one policy to select. To address such a challenge, we propose a new, stronger baseline as follows by defining a $\widetilde{\Pi}^{\mathrm{baseline}} = \{\text{ATLA-PPO}, \text{PA-ATLA-PPO}, \text{WocaR-PPO}\}$. Note that since $\widetilde{\Pi}^{\mathrm{baseline}}$ has effectively aggregated all previous baselines, it should be no worse than them. Now $\widetilde{\Pi}^{\mathrm{baseline}}$ and our $\widetilde{\Pi}$ are comparable since they both use Algorithm 1 to utilize the online reward feedback. The detailed comparison is in Table 3. We can see that even with this new, stronger baseline utilizing the reward feedback in the same way as us, our results are still consistently better.

| Environment | Model | Natural Reward | Random | RS | SA-RL | PA-AD |
|---|---|---|---|---|---|---|
| **Hopper** | $\widetilde{\Pi}^{\text{baseline}}$ | $3624 \pm 186$ | $3605 \pm 41$ | $3284 \pm 249$ | $2442 \pm 150$ | $2627 \pm 254$ |
| | **Ours** | $\mathbf{3652 \pm 108}$ | $\mathbf{3653 \pm 57}$ | $\mathbf{3332 \pm 713}$ | $\mathbf{2526 \pm 682}$ | $\mathbf{2896 \pm 723}$ |
| **Walker2d** | $\widetilde{\Pi}^{\text{baseline}}$ | $4193 \pm 508$ | $4256 \pm 177$ | $4121 \pm 251$ | $4069 \pm 397$ | $3158 \pm 197$ |
| | **Ours** | $\mathbf{6319 \pm 31}$ | $\mathbf{6309 \pm 36}$ | $\mathbf{5916 \pm 790}$ | $\mathbf{6085 \pm 620}$ | $\mathbf{5803 \pm 857}$ |
| **Halfcheetah** | $\widetilde{\Pi}^{\text{baseline}}$ | $6294 \pm 203$ | $6213 \pm 245$ | $5310 \pm 185$ | $\mathbf{5369 \pm 61}$ | $4328 \pm 239$ |
| | **Ours** | $\mathbf{7095 \pm 88}$ | $\mathbf{6297 \pm 471}$ | $\mathbf{5457 \pm 385}$ | $5089 \pm 86$ | $\mathbf{4411 \pm 718}$ |
| **Ant** | $\widetilde{\Pi}^{\text{baseline}}$ | $5617 \pm 174$ | $5569 \pm 132$ | $4347 \pm 170$ | $3889 \pm 142$ | $3246 \pm 303$ |
| | **Ours** | $\mathbf{5769 \pm 290}$ | $\mathbf{5630 \pm 146}$ | $\mathbf{4683 \pm 561}$ | $\mathbf{4524 \pm 79}$ | $\mathbf{4312 \pm 281}$ |

Table 3: Average episode rewards $\pm$ standard deviation over 50 episodes with three baselines on Hopper, Walker2d, Halfcheetah, and Ant. Here $\widetilde{\Pi}^{\text{baseline}}$ is used as a baseline policy class for online adaptation.

This justifies that it is our novel Algorithm 2 for discovering a set of high-quality policies $\widetilde{\Pi}$ that makes ours improve over baselines.

