# OpenReview forum: "Beyond Worst-case Attacks: Robust RL with Adaptive Defense via Non-dominated Policies"
_ICLR.cc/2024/Conference — ICLR 2024 spotlight_

### Official Review · Reviewer_xB8p · 2023-10-29

**Soundness:** 3 good
**Presentation:** 2 fair
**Contribution:** 2 fair
**Rating:** 6
**Confidence:** 3

**Summary:**

The authors consider the problem of synthesising policies for reinforcement learning (RL) that are both robust to adversarial perturbations and accurate to natural data. In particular, the authors propose an online adaptive framework that learns a finite set of policies that is used to ensure protection to adversarial attacks. The effectiveness of the approach of the authors is investigated on several standard RL benchmarks.

**Strengths:**

- Finding robust policies that are not overly conservative is an important problem and of interest for the ICLR community

- The approach proposed by the authors is promising and shows empirical improvements compared to state of the art.

- The authors provide bounds on the regret of the proposed algorithm

**Weaknesses:**

- The paper is not well written. There are typos, some notation is not defined, and some parts are not very clear. Some examples:

a) in the definition of J(\pi,v) ,in the first expectation, it is missing where z is sampled from.
b) \pi^tt is not defined (I guess you missed {}, but the same notation appears multiple times)
c)The update rule for \omega is not explained in the main text
d) in the abstract and intro the authors talk about "the importance of a finite and compact \mathcal{\PI}", but any finite set is also compact.
e) k in line 6 of Algorithm 2 not defined

- In order to talk about optimality, convergence guarantees, and scalability, in Theorem 4.10, it would be important that the authors report an upper bound of K.

- I am confused by the result in Table 1. From the results it seems that the proposed approach obtains higher rewards against worst case perturbations compared to methods explicitly trained against worst case perturbations. Am I missed something? In any case, a discussion on the results and about their interpretation should be given

**Questions:**

See Weaknesses Section

---

> ### Author Response · Authors · 2023-11-18
> **Response to Reviewer xB8p**
>
> We thank Reviewer xB8p for the detailed comment and insightful feedback. We are encouraged the reviewer finds that our approach is "promising" with "empirical improvements". We address Reviewer xB8p's concerns and questions below:
>
> ---
> ### [1/1] Response to weakness
>
> ---
> > Q1: The paper is not well written. There are typos, some notation is not defined, and some parts are not very clear. Some examples: a) in the definition of J(\pi,v) ,in the first expectation, it is missing where z is sampled from. b) \pi^tt is not defined (I guess you missed {}, but the same notation appears multiple times) c)The update rule for \omega is not explained in the main text d) in the abstract and intro the authors talk about "the importance of a finite and compact \mathcal{\PI}", but any finite set is also compact. e) k in line 6 of Algorithm 2 not defined
>
> We apologize for the potential confusion and have revised our draft accordingly.
>
> **a)** In the last paragraph of Section 3, we have pointed out that $z$ is sampled from a fixed distribution $\mathbb{P}(z)$, which can be Gaussian or uniform. This distribution is just used to randomize the victim policy and does not affect our algorithm framework. Therefore, in the definition of $J(\pi, \nu)$ we omit the distribution of $z$ and will make it more clear in the revision.
>
> **b)** $\pi^tt$ (\pi^tt) is indeed a typo. It should be $\\{\pi^t\\}\_{t\in[T]}$
>
> **c)** The update of $\omega$ follows the standard adversarial bandit algorithm, EXP3. Essentially, during Algorithm 1, $\omega$ gradually adjusts the probability of selecting each arm according to the observed rewards for each arm in an exponential fashion. Therefore, we did not describe it intuitively in the main text due to the page limit.
>
> **d)** We are sorry for the misuse of "compact". Here by being "compact", we do not mean the mathematical definition of compactness for a set in real analysis. Instead, we just want to convey the intuition that $\tilde{\Pi}$ should not contain redundant policies, which we discussed in depth in Section 4. Therefore, we shall replace "finite and compact $\tilde{\Pi}$" by "small and finite $\tilde{\Pi}$".
>
> **e)** We believe $k$ has been defined in line 3 of Algorithm 2, indicating what the current iteration is.
>
> Again, we thank and accept all the suggestions from the reviewer. We have revised our paper accordingly.
>
>
> ---
> > Q2: In order to talk about optimality, convergence guarantees, and scalability, in Theorem 4.10, it would be important that the authors report an upper bound of K.
>
> We thank the reviewer for bringing this up. The reason why we do not discuss the upper bound of $K$ is as follows
> - **A useful upper bound should be problem-dependent and thus highly non-trivial to analyze. Instead, we choose to give more qualitative intuitions of whether $K$ will be large or small.** Quantitatively, such an upper bound should be ideally related to many factors, such as the attack budget, Lipschitzness of the transition, and rewards as pointed out by Remark 4.4., etc. Therefore, we find it is an extremely complicated problem to give a precise relationship between the upper bound and those problem-dependent quantities. But intuitively, we do know lower attack budget or Lipschitzness indicates a smaller $K$.
> - **A relatively direct and coarse upper bound will be the size of all deterministic policies.** This is because any (potentially random) policy can be expressed as a mixture of deterministic policies by Lemma B.1. Therefore, the deterministic policy set incurs $0$ optimality gap. Meanwhile, we also know our algorithm can find the smallest policy set that incurs $0$ optimality gap. Thus, the size of $\tilde{\Pi}$ will always be upper bound by the size of deterministic policy space.
> - **The convergence guarantee means our method always converges to the policy set incurring $0$ optimality gap, while scalability means it always finds such a policy set of the smallest size.** Therefore, our method is the most efficient in terms of identifying such a desirable policy space. The exact size of the smallest non-dominated policy set is determined by the problem nature instead of our algorithm.

---

> ### Author Response · Authors · 2023-11-18
> **Response to Reviewer xB8p**
>
> > Q3: I am confused by the result in Table 1. From the results it seems that the proposed approach obtains higher rewards against worst case perturbations compared to methods explicitly trained against worst case perturbations. Am I missed something? In any case, a discussion on the results and about their interpretation should be given
>
> We do agree with the reviewer that further discussions on the improvement against worse-case attacks are valuable.
>
> **The reason why we also achieve improved robustness against worst-case attackers is due to the limited algorithmic efficiency and instability of baselines**. It is worth pointing out that all related works [1, 2, 3] share the same objective of maximizing the worst-case robustness. However, in terms of empirical performance, [2] improves over [1] and [3] improves over [2]. Therefore, it is highly likely that existing approaches may not find the global optimal solution reliably. Specifically, one possible explanation from the perspective of optimization is that baselines like [1, 2, 3] could often converge to a local optimum, while our approach **explicitly encourages the new policy to behave differently in the reward space** compared with policies discovered already, thus not stuck at a single local optimal solution easily. To further verify our hypothesis, we design the experiments as follows.
>
> Firstly, note the number we report in Table 1 is a medium number of rewards for **$21$ victims agents** following the setup of related works [1, 2, 3]. To verify our hypothesis, here we compare the performance of the best victim and the medium victim of different approaches.
>
> | Environments  | Model       | PA-AD (against medium victim) | PA-AD (against best victim) |
> | -----------   | ----------- | -------------- | --------------- |
> | Hopper        | PA-ATLA-PPO | $2521\pm325$   | $3129\pm316$    |
> |               | WocaR-PPO   | $2579\pm229$   | $3284\pm193$    |
> |               | Ours        | $2896\pm723$   | $3057\pm809$    |
> | Walker2d      | PA-ATLA-PPO | $2248\pm131$   | $3561\pm357$    |
> |               | WocaR-PPO   | $2722\pm173$   | $4239\pm295$    |
> |               | Ours        | $5803\pm857$   | $6052\pm944$    |
> | Halfcheetah   | PA-ATLA-PPO | $3840\pm273$   | $4260\pm193$    |
> |               | WocaR-PPO   | $4269\pm172$   | $4579\pm368$    |
> |               | Ours        | $4411\pm718$   | $4533\pm692$    |
> | Ant           | PA-ATLA-PPO | $2986\pm364$   | $3529\pm782$    |
> |               | WocaR-PPO   | $3164\pm163$   | $4273\pm530$    |
> |               | Ours        | $4312\pm281$   | $4406\pm329$    |
>
> **We notice that baselines like [2, 3] can match the high performance of ours in terms of the best run, while the median is low by a large margin.** This shows baseline methods can achieve high worst-case robustness occasionally as its objective encourages so explicitly but not reliably. In contrast, our methods are much more stable. In conclusion, we believe there is room for improvement regarding worst-case robustness even if [1, 2, 3] explicitly maximizes it.
>
>
> ---
>
> We greatly appreciate Reviewer xB8p's valuable feedback and constructive suggestions. We are happy to answer any further questions.
>
> Paper6801 Authors
>
> ---
>
> [1] Zhang, Huan, et al. "Robust reinforcement learning on state observations with learned optimal adversary." arXiv preprint arXiv:2101.08452 (2021).
>
> [2] Sun, Yanchao, et al. "Who is the strongest enemy? towards optimal and efficient evasion attacks in deep rl." arXiv preprint arXiv:2106.05087 (2021).
>
> [3] Liang, Yongyuan, et al. "Efficient adversarial training without attacking: Worst-case-aware robust reinforcement learning." Advances in Neural Information Processing Systems 35 (2022): 22547-22561.

---

> ### Author Response · Authors · 2023-11-20
> **Does our response address your concerns?**
>
> Dear reviewer xB8p,
>
> As the stage of the review discussion is ending soon, we would like to kindly ask you to review our revised paper as well as our response and consider making adjustments to the scores.
>
> Besides correcting all typos according to your valuable suggestions, we would like to point out that:
>
> (a) We have addressed the concern about the upper bound of $K$ by giving a comprehensive discussion in our first response.
>
> (b) We have addressed the concern about the result on worst-case perturbations by providing our interpretations with supplementary experimental results in our second response.
>
> Please let us know if there are any other questions. We would appreciate the opportunity to engage further if needed.
>
> Best regards,
>
> Paper6801 Authors

---

> > ### Author Response · Authors · 2023-11-22
> > **Is there anything else we can address?**
> >
> > As the review-author phase is ending soon, we would like to know whether we have addressed your concerns? If there is anything else we can do to help you better evaluate our paper, please do not hesitate to let us know!
> >
> > Best regards,
> >
> > Paper6801 Authors

---

### Official Review · Reviewer_t6xM · 2023-10-30

**Soundness:** 3 good
**Presentation:** 3 good
**Contribution:** 3 good
**Rating:** 8
**Confidence:** 2

**Summary:**

This paper introduces the test-time adaptation idea to adversarial robustness of RL models. More specifically at training time instead of identifying a single fixed policy, the learner maintains a set of policies, and then at the test time, it adapts to a particular trajectory and then chooses a good policy to react.

The authors have also considered how to attack such adaptation defense in a clear way. And various experiments have demonstrated the superiosity of the framework.

**Strengths:**

A novel and natural idea to introduce test-time adaptation to the RL. The authors have done a great job in carefully considering the security model, and discussed in detail how to attack such a test-time adaptation framework.

The authors have also discussed in details both theoretical and empirical evidence of the proposed framework, and it seems that this work can improve state of the art by a large margin.

Overall I found this paper a really nice to read.

**Weaknesses:**

The part about adaptive attacks seem somewhat weak -- namely, if the adversary has more knowledge about the defender, what can happen? The paper does not really seem to have clearly formulated what adaptivity really means in this case. However, I think this could be a future direction, given that this paper already has some novel ideas.

**Questions:**

No specific questions, my main concerns have been stated above.

---

> ### Author Response · Authors · 2023-11-18
> **Response to Reviewer t6xM**
>
> We thank Reviewer t6xM for the detailed comment and insightful feedback. We are encouraged the reviewer finds our paper introduces "a novel and natural idea" and discusses "in detail both theoretical and empirical evidence". We address Reviewer t6xM's concerns and questions below:
> ### [1/1] Response to weakness
> > Q1: The part about adaptive attacks seem somewhat weak -- namely, if the adversary has more knowledge about the defender, what can happen? The paper does not really seem to have clearly formulated what adaptivity really means in this case. However, we think this could be a future direction, given that this paper already has some novel ideas.
> - **If the adversary has more knowledge, it could develop a more effective attacking strategy.** If the attacker knows $\tilde{\Pi}$ in advance, it could attack each policy in $\tilde{\Pi}$ and forms a corresponding $\tilde{\Pi}^{\text{attack}}$. With $\tilde{\Pi}^{\text{attack}}$, the attacker could also run a bandit-type algorithm to adapt to the victim.
> - **By adaptive attack in our paper, we refer to the general scenario where attackers can change their policy over time.** We use those heuristic adaptive attackers to model the practical scenarios of attacks beyond the worst cases, helping us to better evaluate our approach.
>
> In general, we believe understanding the optimal adaptive strategy from the attacker's perspective is an important problem and we will leave it as a future work.
>
> ---
>
> We greatly appreciate Reviewer t6xM's valuable feedback and constructive suggestions. We are happy to answer any further questions.
>
>
> Paper6801 Authors

---

### Official Review · Reviewer_M1op · 2023-10-31

**Soundness:** 3 good
**Presentation:** 3 good
**Contribution:** 4 excellent
**Rating:** 8
**Confidence:** 2

**Summary:**

This paper proposes an approach to learning robust RL policies that improve robustness in the case of no attacks or weak attacks while still maintaining robustness against worst-case attacks. This approach entails adaptively picking policies at each time step from a fixed policy class in order to minimize a particular notion of regret. The fixed policy class is constructed during training with the aim of minimizing its size while minimizing regret-related optimality gaps compared to using the full policy class - notably, via discovery of non-dominated policies. The authors provide theoretical results on the hardness of the problem, optimality gaps between the fixed (refined) policy class and the full policy class, the regret of their algorithm, and the worst-case size of the refined policy class. They demonstrate empirical performance on four Mujoco environments with continuous action spaces (Hopper, Walker2d, Halfcheetah, and Ant) and show strictly improved performance against baselines across no-attack, non-worst-case-attack, and worst-case-attack scenarios.

**Strengths:**

Formulation: The authors formulate an interesting and important task of improving RL performance against non-worst-case state-adversarial attacks, in addition to no attacks and worst-case state-adversarial attacks. They theoretically establish its difficulty, opening up an interesting line of future research.

Theoretical results: While I did not check the proofs in detail, the authors seemingly present useful theoretical results regarding optimality gaps presented by the use of non-infinite policy classes, as well as the performance of their algorithm.

Empirical results: The empirical performance is strong, with comparison on several environments against several baselines and attack strategies.

**Weaknesses:**

Questions regarding definition of regret (Definition 4.2): It is unclear to me why the notion of regret does not allow $\pi \in \Pi$ to be picked adaptively at each time step in the same way that $\pi^t \in \Pi$ is picked adaptively. While this may not matter in the case of an infinite $\Pi$ given that the adaptive policy is likely contained within $\Pi$, once $\Pi$ is restricted to $\tilde{Pi}$ with finite cardinality, this may no longer be the case. In other words, I wonder whether the notion of regret used in this work may be too weak.

Questions regarding experimental results: While the aim of the paper is stated as improving performance against no attacks/non-worst-case adversarial attacks while *maintaining* performance against worst-case adversarial results, the empirical results show the proposed method *exceeding* performance against worst-case adversarial attacks compared to baselines (in some cases, substantially - see e.g. Walker2d vs. PA-AD). This seems to contradict "no free lunch" intuitions. The authors should further explain why we are seeing this behavior in the experiments (potentially through the use of ablation studies), and whether there are any tradeoffs.

Minor points:
* Section 3: $H$ does not seem to be defined
* Section 3: Typo - "discounted factor" --> "discount factor"
* Proposition 4.3: How does the choice of $\alpha$ relate to the rest of the problem setup?

**Questions:**

* Why is the notion of regret that is defined not overly weak?
* Why are the experimental results not too good to be true? Are there any tradeoffs inherent in the method or results?

---

> ### Author Response · Authors · 2023-11-18
> **Response to Reviewer M1op**
>
> We thank Reviewer M1op for the detailed comment and insightful feedback. We are encouraged the reviewer finds that we formulate "an interesting and important task", and present "useful theoretical results" and "strong empirical performance". We address Reviewer M1op's concerns and questions below:
>
> ---
> ### [1/2] Response to weakness
>
> ---
> > Q1: Questions regarding definition of regret (Definition 4.2): It is unclear to me why the notion of regret does not allow $\pi\in\Pi$ to be picked adaptively at each time step in the same way that $\pi^t\in\Pi$ is picked adaptively. While this may not matter in the case of an infinite $\Pi$ given that the adaptive policy is likely contained within $\Pi$, once $\Pi$ is restricted to $\tilde{\Pi}$ with finite cardinality, this may no longer be the case. In other words, I wonder whether the notion of regret used in this work may be too weak.
>
> We thank the reviewer for bringing up such an important and insightful question.
>
> - **Our regret definition is of the most standard form and naturally improves related works aiming for a single static robust policy.** Our Definition 4.2 follows the most standard way of defining regret by comparing it with the **best fixed policy in hindsight**. Therefore, it is a commonly accepted notion for online adaptation and serves as the first step for studying other stronger notions. In addition, since all existing works aim for a single static robust policy, it is also quite natural for us to compare with the best fixed policy in our definition of regret.
>
> - **The regret notion proposed by the reviewer is also compatible with our algorithm framework.** Indeed, as what the reviewer describes, when the comparison policy $\pi$ is also picked adaptively in Definition 4.2, this leads to an important variant of regret called **dynamic regret**. To be concrete, it is defined as follows
> $$
> \text{D-Regret}(T):=\sum_{t=1}^{T}\max_{\pi\in\Pi}J(\pi, \nu^t) - J(\pi^t, \nu^t)
> $$
> Compared with the regret defined in our paper as
> $$
> \text{Regret}(T):=\max_{\pi\in\Pi}\sum_{t=1}^{T}J(\pi, \nu^t) - J(\pi^t, \nu^t),
> $$
> the dynamic regret always holds that $\text{D-Regret}(T)\ge \text{Regret}(T)$. Therefore, low dynamic regret is a stronger guarantee than low regret.
>
> To see how our framework can deal with the dynamic regret:
> 1. **For Algorithm 2 (iterative discovery): It is indifferent to the choice of regret or dynamic regret.** In terms of this new and stronger regret, discovering $\tilde{\Pi}$ with small $\text{Gap}(\tilde{\Pi}, \Pi)$ still suffices since our Proposition 4.7 can be extended in a direct way to dynamic regret:
> $$
> \frac{\operatorname{D-Regret}(T)}{T}\le \frac{\tilde{\operatorname{D-Regret}}(T)}{T} + \operatorname{Gap}(\widetilde{\Pi}, \Pi).
> $$
> Therefore, as long as $\operatorname{Gap}(\tilde{\Pi}, \Pi)$ is small, using $\tilde{\Pi}$ instead of $\Pi$ for online adaptation sacrifices no optimality even in terms of the new stronger regret metric.
> 2. **For Algorithm 1 (online adaptation): To ensure low dynamic regret, the attacker policy is required to change slowly and then Algorithm 1 can be replaced with a dynamic regret minimization bandit algorithm. Otherwise, it is intrinsically hard.** If the attacker policy is changing fast and arbitrarily, even $\tilde{\operatorname{D-Regret}}(T)$ cannot be sublinear [1, 2], let alone $\operatorname{D-Regret}(T)$. However, if the attacker policy is changing slowly, our framework readily works by replacing Algorithm 1 with those algorithms developed for bandit dynamic regret minimization under various types of conditions that rewards of bandit arms are changing slowly, for e.g., [1, 2]. It is worth mentioning that the key algorithmic design for those algorithms is simply restarting Algorithm 1 periodically or maintaining a finite window.

---

> ### Author Response · Authors · 2023-11-18
> **Response to Reviewer M1op**
>
> ---
> > Q2: Questions regarding experimental results: While the aim of the paper is stated as improving performance against no attacks/non-worst-case adversarial attacks while maintaining performance against worst-case adversarial results, the empirical results show the proposed method exceeding performance against worst-case adversarial attacks compared to baselines (in some cases, substantially - see e.g. Walker2d vs. PA-AD). This seems to contradict "no free lunch" intuitions. The authors should further explain why we are seeing this behavior in the experiments (potentially through the use of ablation studies), and whether there are any tradeoffs.
>
> We do agree with the reviewer that further discussions on the improvement against worse-case attacks are valuable.
> - **The experimental results do not contradict "no free lunch" intuition**. The improvement of our approach comes from utilizing the online reward feedback while the baseline approaches neglect such information by deploying a static victim. Therefore, by adapting to the unknown attacker, our approach achieves robustness against both worst-case and non-worst-case attackers. In other words, if we also just deploy a single static victim as in baseline approaches, we agree with the reviewer that there is an inevitable trade-off such that improved robustness against worst-case attackers and non-worst-case attackers might not be achieved at the same time due to the "no free lunch" intuitions.
> - **The reason why our method improves robustness against worst-case attackers is due to the limited algorithmic efficiency and instability of baselines**. It is worth pointing out that all related works [3, 4, 5] share the same objective of maximizing the worst-case robustness. However, in terms of empirical performance, [4] improves over [3] and [5] improves over [4]. Therefore, it is more likely that existing approaches do not find the global optimal solution reliably or efficiently. One possible explanation from the perspective of optimization is that baselines like [3, 4, 5] could often converge to a local optimum, while our approach explicitly encourages the new policy to behave differently in the reward space compared with policies discovered already, thus not stuck at a single local optimal solution easily. To further verify such a hypothesis, we design the experiments as follows. Note the number we report in Table 1 is a medium reward of **$21$ trained victim agents** following the setup of related works [3, 4, 5]. Therefore, here we report the performance of the best one and the median of different approaches.
>
> | Environments  | Model       | PA-AD (against medium victim) | PA-AD (against best victim) |
> | -----------   | ----------- | -------------- | --------------- |
> | Hopper        | PA-ATLA-PPO | $2521\pm325$   | $3129\pm316$    |
> |               | WocaR-PPO   | $2579\pm229$   | $3284\pm193$    |
> |               | Ours        | $2896\pm723$   | $3057\pm809$    |
> | Walker2d      | PA-ATLA-PPO | $2248\pm131$   | $3561\pm357$    |
> |               | WocaR-PPO   | $2722\pm173$   | $4239\pm295$    |
> |               | Ours        | $5803\pm857$   | $6052\pm944$    |
> | Halfcheetah   | PA-ATLA-PPO | $3840\pm273$   | $4260\pm193$    |
> |               | WocaR-PPO   | $4269\pm172$   | $4579\pm368$    |
> |               | Ours        | $4411\pm718$   | $4533\pm692$    |
> | Ant           | PA-ATLA-PPO | $2986\pm364$   | $3529\pm782$    |
> |               | WocaR-PPO   | $3164\pm163$   | $4273\pm530$    |
> |               | Ours        | $4312\pm281$   | $4406\pm329$    |
>
> - **We notice that baselines like [4, 5] can almost match our method's high performance in terms of the best run, while the average or median is low by a large margin**. This means baselines can also achieve strong worst-case robustness occasionally as its objective explicitly encourages so, but not reliably. In contrast, our methods are much more stable. In conclusion, we believe there is room for improvement regarding worst-case robustness even if [3, 4, 5] maximizes it explicitly.
>
> ---
> > Q3: Section 3: $H$ does not seem to be defined
>
> $H$ here indicates the horizon of the MDP, which we describe in the second paragraph of Section 3 for the first time. We shall clarify the meaning of $H$ more clearly in our revision.
>
> ---
> > Q4: Section 3: Typo - "discounted factor" --> "discount factor"
>
> We thank the suggestion by the reviewer and will revise accordingly.
>
> ---
> > Q5: Proposition 4.3: How does the choice of $\alpha$ relate to the rest of the problem setup?
>
> $\alpha$ indicates the growth of the regret. **We aim to develop an algorithm with the corresponding $\alpha<1$ so that $\lim_{T\rightarrow +\infty}\frac{\text{Regret}(T)}{T}= 0$, i.e., the policy sequence $\\{\pi^t\\}_{t\in[T]}$ is asymptotically optimal.** In Proposition 4.5 & 4.7, the corresponding $\alpha$ is $\frac{1}{2}$.

---

> > ### Author Response · Authors · 2023-11-18
> > **Response to Reviewer M1op**
> >
> > ### [2/2] Response to questions
> >
> > ---
> > > Q6: Why is the notion of regret that is defined not overly weak?
> >
> > As in our response to Q1, our regret is not weak but standard and naturally improves over existing works. Meanwhile, our framework also readily adapts to stronger regret notions like dynamic regret.
> >
> > ---
> > > Q7: Why are the experimental results not too good to be true? Are there any tradeoffs inherent in the method or results?
> >
> > As in our response to Q2, the improvement in worst-case robustness comes from the limited algorithmic efficiency and instability of baselines. One trade-off we realize is between exploration and exploitation during online adaptation. If the online adaptation stage is too short for Algorithm 1 to identify a reliable mixture policy, simply deploying a single static robust policy like [3, 4, 5] could potentially outperform our method.
> >
> > ---
> >
> > We greatly appreciate Reviewer M1op's valuable feedback and constructive suggestions. We are happy to answer any further questions.
> >
> > Paper6801 Authors
> >
> > ---
> > [1] Besbes, Omar, Yonatan Gur, and Assaf Zeevi. "Stochastic multi-armed-bandit problem with non-stationary rewards." Advances in neural information processing systems 27 (2014).
> >
> > [2] Cheung, Wang Chi, David Simchi-Levi, and Ruihao Zhu. "Learning to optimize under non-stationarity." The 22nd International Conference on Artificial Intelligence and Statistics. PMLR, 2019.
> >
> > [3] Zhang, Huan, et al. "Robust reinforcement learning on state observations with learned optimal adversary." arXiv preprint arXiv:2101.08452 (2021).
> >
> > [4] Sun, Yanchao, et al. "Who is the strongest enemy? towards optimal and efficient evasion attacks in deep rl." arXiv preprint arXiv:2106.05087 (2021).
> >
> > [5] Liang, Yongyuan, et al. "Efficient adversarial training without attacking: Worst-case-aware robust reinforcement learning." Advances in Neural Information Processing Systems 35 (2022): 22547-22561.

---

> ### Author Response · Authors · 2023-11-20
> **Does our response address your concerns?**
>
> Dear reviewer M1op,
>
> As the stage of the review discussion is ending soon, we would like to kindly ask you to review our revised paper as well as our response and consider making adjustments to the scores.
>
> Besides correcting all typos according to your valuable suggestions, we would like to point out that:
>
> (a) We have addressed the concern that **the notion of regret used in this work may be too weak** by providing a comprehensive analysis in our first response, and
>
> (b) We have addressed the concern about the result on worst-case perturbations by providing our interpretations with supplementary experimental results in our second response.
>
> Please let us know if there are any other questions. We would appreciate the opportunity to engage further if needed.
>
> Best regards,
>
> Paper6801 Authors

---

> > ### Comment · Reviewer_M1op · 2023-11-21
> > **Response to authors**
> >
> > Thank you to the authors for the response and revisions. Please see my responses below.
> >
> > **Major outstanding issue – contribution and experimental results:** I appreciate the authors’ clarifications regarding the strength of their results coming from using online reward feedback vs. a static victim. However, this means that the paper is making two major changes compared to the baselines: (a) updated notion of robustness beyond just worst-case attacks, (b) use of online reward feedback. Without further experiments, it is not possible to distinguish which of the contributions is actually making the most substantive difference. While the new experiment perhaps provides some hints (online reward feedback may be allowing escape from local optima), more experiments/ablation studies are needed to fully identify the source of improvement (as well as a clearer explanation of the current experiment -- I'm not sure I fully understand the experimental setup from the description provided).
> >
> > **Minor point -- the notion of regret:** I appreciate the authors’ clarifications regarding using an established notion of regret, rather than redefining it. However, I still do believe it is the case that restricting the policy set $\Pi$ to $\tilde{\Pi}$ significantly changes even the existing notion of regret in a way that is salient to the present work. Specifically, for an infinite $\Pi$, the existing notion of regret and the notion of dynamic regret are the same, since any dynamic policy should be contained in $\Pi$. Once $\Pi$ is restricted to the finite $\tilde{\Pi}$ , this is no longer the case. More discussion of this could be helpful.

---

> ### Author Response · Authors · 2023-11-22
> **Thanks for your reply!**
>
> Thank you very much for your further response and for raising discussion-worthy questions. We would like to provide more discussions as follows
>
> ---
> > Q1. Major outstanding issue – contribution and experimental results: I appreciate the authors’ clarifications regarding the strength of their results coming from using online reward feedback vs. a static victim. However, this means that the paper is making two major changes compared to the baselines: (a) updated notion of robustness beyond just worst-case attacks, (b) use of online reward feedback. Without further experiments, it is not possible to distinguish which of the contributions is actually making the most substantive difference. While the new experiment perhaps provides some hints (online reward feedback may be allowing escape from local optima), more experiments/ablation studies are needed to fully identify the source of improvement (as well as a clearer explanation of the current experiment -- I'm not sure I fully understand the experimental setup from the description provided).
>
> ### **Our updated robustness notion v.s. using online feedback**:
> Firstly, we believe
> "*(a) updated notion of robustness beyond just worst-case attacks*" motivates Algorithm 2 (iterative discovery) to minimize $\text{Gap}(\tilde{\Pi}, \Pi)$.
>
> "*(b) use of online reward feedback*" corresponds to Algorithm 1 (online adaptation).
>
> To clarify (a), i.e., our novel Algorithm 2 to minimize $\text{Gap}(\tilde{\Pi}, \Pi)$ is our major contributions and novelties instead of using online reward feedback, we adopt the reviewer's suggestions for further ablation studies **by also allowing baselines to use online reward feedbacks through our Algorithm 1**.
>
> However, all baselines to our knowledge are essentially a **single** victim policy instead of a set, making it trivial to run Algorithm 1 since it will only have one policy to select.
>
> To address such a challenge, we propose a new, stronger baseline as follows by defining a $\tilde{\Pi}^{\text{baseline}}=\\{\text{ATLA-PPO}, \text{PA-ATLA-PPO}, \text{WocaR-PPO}\\}$. Note that since $\tilde{\Pi}^{\text{baseline}}$ has effectively aggregated all previous baselines, it should be no worse than them. Now $\tilde{\Pi}^{\text{baseline}}$ and our $\tilde{\Pi}$ are comparable since they **BOTH** use Algorithm 1 to utilize the online reward feedback. The detailed comparison is in the following table
>
> | Environments  | Model       | Natural Reward | Random          | RS              | SA-RL           | PA-AD           |
> | -----------   | ----------- | -------------- | --------------- | --------------- | --------------- | --------------- |
> | Hopper        | $\tilde{\Pi}^{\text{baseline}}$ | $3624\pm186$ | $3605\pm41$ | $3284\pm249$ | $2442\pm150$ | $2627\pm254$ |
> |               | Ours        | $3652\pm108$ | $3653\pm57$ | $3332\pm713$ | $2526\pm682$ | $2896\pm723$ |
> | Ant           | $\tilde{\Pi}^{\text{baseline}}$ | $5617\pm174$ | $5569\pm132$ | $4347\pm170$ | $3889\pm142$ | $3246\pm303$ |
> |               | Ours        | $5769\pm290$ | $5630\pm146$ | $4683\pm561$ | $4524\pm79$ | $4312\pm281$ |
>
> ***Due to time constraints, we currently only have experimental results for two environments and will provide a more complete evaluation as soon as possible.***
>
> We can see that even with this new, stronger baseline utilizing the reward feedback in the same way as us, our results are still consistently better. This justifies that it is our novel Algorithm 2 for discovering a set of high-quality policies $\tilde{\Pi}$ that makes ours improve over baselines.
>
> **On the other hand, we believe Algorithm 1 itself, proposing to utilize the online reward feedback to adapt to different dynamic attacker is already a significant contribution compared existing papers.**
>
> ### **Further clarifications for the improvement of worst-case attacks**
>
> We believe there might be a little bit of misunderstanding on our original responses to clarify the improvement on worst-case attacks.
>
> When we talked about "escaping from local optima", we meant the "local optima" of the **training** algorithm, like PA-ATLA-PPO, WocaR-PPO, and our Algorithm 2. Those algorithms happen before online adaptation, and online reward feedback has not been used yet. During online adaptation, since policies discovered are fixed and only the probability of sampling each policy changes, there are no local optima in this stage. Therefore, we believe the advantage of our approach comes from Algorithm 2, the iterative discovery that explicitly encourages victim policies to behave differently in the reward space.

---

> ### Author Response · Authors · 2023-11-22
> **Thanks for your reply!**
>
> > Q2. Minor point -- the notion of regret: I appreciate the authors’ clarifications regarding using an established notion of regret, rather than redefining it. However, I still do believe it is the case that restricting the policy set $\Pi$ to $\tilde{\Pi}$ significantly changes even the existing notion of regret in a way that is salient to the present work. Specifically, for an infinite $\Pi$, the existing notion of regret and the notion of dynamic regret are the same, since any dynamic policy should be contained in $\Pi$. Once  $\Pi$ is restricted to the finite
> $\tilde{\Pi}$, this is no longer the case. More discussion of this could be helpful.
> ### **More discussions on dynamic regret**
> Firstly, we thank the reviewer's appreciating of us using the established notion of regret. We would like to provide more mathematical discussions on regret and dynamic regret. We let
> $$\pi^*\in\mathop{\arg\max}\_{\pi\in\Pi}\sum\_{t=1}^T J(\pi,\nu^t)-J(\pi^t,\nu^t)$$
>
> Thus, we have the following inequalities
> $$
> \begin{aligned}
> \text{Regret}(T)=&\max_{\pi\in\Pi}\sum_{t=1}^T J(\pi,\nu^t)-J(\pi^t,\nu^t)\\\\
> =&\sum_{t=1}^T J(\pi^*,\nu^t)-J(\pi^t,\nu^t)\\\\
> \overset{(a)}{\leq}&\sum_{t=1}^T \max_{\pi\in\Pi}J(\pi,\nu^t)-J(\pi^t,\nu^t)\\\\
> =&\text{D-Regret}(T)
> \end{aligned}
> $$
> where $(a)$ is due to $J(\pi^*,\nu^t)\leq\max_{\pi\in\Pi}J(\pi,\nu^t)$. Therefore, the two notions are equivalent when for any $t$: $J(\pi^*,\nu^t)=\max_{\pi\in\Pi}J(\pi,\nu^t)$.
>
> ### **Effects of using $\tilde{\Pi}$ for dynamic regret**
> Firstly, we believe what the reviewer highlights is that
> $$
> \text{D-Regret}(T)-\text{Regret}(T)\le \tilde{\text{D-Regret}}(T)-\tilde{\text{Regret}}(T)
> $$
> In other words, using $\tilde{\Pi}$ may result in a larger gap between dynamic regret and regret. We agree with the reviewer's intuition. Meanwhile, we prove an upper bound on $\tilde{\text{D-Regret}}(T)-\tilde{\text{Regret}}(T)$ here. We note the relationship that
> $$
> \tilde{\text{D-Regret}}(T)\le \text{D-Regret}(T),
> $$
>
> $$
> \tilde{\text{Regret}}(T)\ge \text{Regret}(T) -T\text{Gap}(\tilde{\Pi}, \Pi),
> $$
> Plugging them into $\tilde{\text{D-Regret}}(T)-\tilde{\text{Regret}}(T)$ we get the following relationship
> $$
> \tilde{\text{D-Regret}}(T)-\tilde{\text{Regret}}(T)\le \text{D-Regret}(T)-\text{Regret}(T)+T\text{Gap}(\tilde{\Pi}, \Pi)
> $$
>
> This inequality shows that the gap between dynamic regret and regret might increase but not a lot as long as $\text{Gap}(\tilde{\Pi}, \Pi)$ is small, which is the motivation of Algorithm 2.
>
>
> ---
> We hope our response further addresses your concerns and helps you understand the contribution of our work to the robust RL community. We welcome any further discussion.
>
> Best regards,
>
> Paper6801 Authors

---

> ### Author Response · Authors · 2023-11-22
> **Additional response**
>
> We here update the results of the new ablation study to justify our improvement does come from Algorithm 2, iterative discovery, instead of just using online feedback. Here are the results for the other two environments.
>
> | Environments  | Model       | Natural Reward | Random          | RS              | SA-RL           | PA-AD           |
> | -----------   | ----------- | -------------- | --------------- | --------------- | --------------- | --------------- |
> | Walker2d      | $\tilde{\Pi}^{\text{baseline}}$ | $4193\pm508$ | $4256\pm177$ | $4121\pm251$ | $4069\pm397$ | $3158\pm197$ |
> |               | Ours        | $6319\pm31$ | $6309\pm36$ | $5916\pm790$ | $6085\pm620$ | $5803\pm857$ |
> | Halfcheetah   | $\tilde{\Pi}^{\text{baseline}}$ | $6294\pm203$ | $6213\pm245$ | $5310\pm185$ | $5369\pm61$ | $4328\pm239$ |
> |               | Ours        | $7095\pm88$ | $6297\pm471$ | $5457\pm385$ | $5096\pm86$ | $4411\pm718$ |
>
> We can see that the conclusion is unchanged (even with this new, stronger baseline utilizing the reward feedback in the same way as us, our results are still consistently better).
>
>
> If there are any additional concerns, please do not hesitate to let us know. We believe it shall greatly improve our paper!
>
> Paper6801 Authors

---

### Official Review · Reviewer_F1uH · 2023-11-02

**Soundness:** 3 good
**Presentation:** 2 fair
**Contribution:** 3 good
**Rating:** 6
**Confidence:** 2

**Summary:**

This paper proposes a method to address performance loss of robust reinforcement learning under weak or no adversarial attacks. To be more specific, they propose a method called PROTECTED, which first find a suitable finite set $\tilde\Pi$ of non-dominated policie during the training, and them adaptively select the policy at each time in a bandit-like way when testing. Empirical results are presented to backup the method.

**Strengths:**

This paper proposes an interesting approach for robust reinforcement learning by adaptively switch among non-dominated policies. The definition of "non-dominated policy" seems new. The algorithm to discover $\tilde{\Pi}$ is non-trivial and might be considered in future studies.

**Weaknesses:**

Overall, the presentation of the current paper can be improved. I find the settings in Section 4 confusing. Specifically, please consider the followings.
1. While the authors define a discount factor $\gamma$ in Section 3 (which seems to be used in an infinite horizon MDP), it is not used in the rest of the paper, where a MDP with finite horizon $H$ is considered. Can the result be extended to infinite horizon?
2. The authors propose the new regret in equation 4.1 that is based on the value function rather than reward. I feel it is not fully discussed why such a regret is appealing and how is it related with the commonly used reward-based regret.
3. Also in equation 4.1, it seems that the regret depend on the attacker policy $\nu^t$ for each time $t$, where $\nu^t$ can be arbitrary. Is this regret well-defined without further confining $\nu^t$?
4. In the paragraph below equation 4.1, the term $\pi^tt\in[T]$ seems like typo. Should it be {$\pi_t: t\in [T]$} instead?

With above issues (especially 2 & 3), I do not fully understand the setting considered in this paper and am unable to evaluate its theoretical contribution accurately.

Besides, another weakness is pointed out by authors in Theorem 4.11 that suggests the size of $\tilde{\Pi}$ can be very large in some cases that is exponential in time horizon $H$.

**Questions:**

Based on Theorem 4.11, will the method be very likely to fail for infinite horizon MDP?

---

> ### Author Response · Authors · 2023-11-18
> **Response to Reviewer F1uH**
>
> We thank Reviewer F1uH for the detailed comment and insightful feedback. We are encouraged the reviewer finds our paper "proposes an interesting approach for robust reinforcement learning" and a "non-trivial algorithm to discover $\tilde{\Pi}$". We address Reviewer F1uH's concerns and questions below:
>
> ---
> ### [1/2] Response to weakness
> ---
> > Q1: While the authors define a discount factor in Section 3 (which seems to be used in an infinite horizon MDP), it is not used in the rest of the paper, where a MDP with finite horizon  is considered. Can the result be extended to infinite horizon?
>
> We apologize for the typo of mentioning the discount factor $\gamma$ in our draft. Indeed, it is not used for our approach and we will remove that in the revision.
>
> **However, our approach does readily handle the setup of infinite-horizon MDP**. Before diving into the setup of robust RL, we point out that for standard RL there is a clear separation between infinite-horizon MDP and finite-horizon MDP since people care about stationary policies in infinite-horizon MDP and non-stationary policies in finite-horizon MDP. Thus, solutions or approaches for infinite-horizon MDP and finite-horizon MDP may not be adapted to each other easily. Nevertheless, that is not the case for robust RL problems since both our paper and related works using the setup of infinite-horizon MDP need to consider history-dependent policies [1, 2].
>
> **Therefore, our approach readily handles infinite-horizon MDP $M=(\mathcal{S}, \mathcal{A}, \mathbb{T}, \mu_1, r, \gamma)$ by a standard reduction (see one example in Section C.3 of [3]) to an approximate *truncated* finite-horizon MDP** $\hat{M}=(\hat{\mathcal{S}}, \hat{\mathcal{A}}, \hat{\mathbb{T}}, \hat{\mu}_1, \hat{r}, H)$, where
> - $\hat{\mathcal{S}}=\mathcal{S}$, $\hat{\mathcal{A}}=\mathcal{A}$, $\hat{\mathbb{T}}=\mathbb{T}$, $\hat{\mu}_1=\mu_1$
> - $\hat{r}_{h}(s, a) = \gamma^{h}r(s, a)$
> - $H: \gamma^{H}\le \epsilon$, thus, $H=\mathcal{O}(\log_{\gamma}\epsilon)=\mathcal{O}(\frac{\log 1/\epsilon}{1-\gamma})$ (when $\gamma$ is close to $1$)
>
> where $\epsilon>0$ is the custom approximation accuracy. Intuitively, this reduction indicates that we can just sample a trajectory in $M$ and stops after $\mathcal{O}(\frac{\log 1/\epsilon}{1-\gamma})$ finite steps to mimic sampling in $\hat{M}$. In other words, we can apply our approach to $\hat{M}$ to discover $\tilde{\Pi}=\\{\pi^{k}\\}\_{k\in[K]}$ and deploy $\\{\pi^{k}\\}_{k\in[K]}$ in the infinite-horizon $M$ by executing them for the first $H$ steps and take arbitrary actions afterwards. Then due to our choice of $H$ above, $\operatorname{Gap}(\tilde{\Pi}, \Pi)$ for the true $M$ will only increase by $\mathcal{O}(\epsilon)$ compared with $\operatorname{Gap}(\tilde{\Pi}, \Pi)$ in the approximated $\hat{M}$. Meanwhile, this reduction is efficient since $H$ only depends logarithmically on $\frac{1}{\epsilon}$.
>
> **Finally, we remark that such truncation is also used in the empirical implementations of [1, 2]** even if they use the setup of infinite-horizon MDP since one can only sample finite steps for an episode.
>
> ---
> > Q2: The authors propose the new regret in equation 4.1 that is based on the value function rather than reward. I feel it is not fully discussed why such a regret is appealing and how is it related to the commonly used reward-based regret.
>
> **Firstly, our regret definition follows the most standard definition for RL problems, which is indeed based on the value function, e.g. [5, 6, 7, 8].** We agree that the definition of regret is usually related to rewards. However, even in the most classical bandit setting, regret is defined w.r.t the **expected reward** of each arm (e.g. Section 4 of [4]) while the exact reward feedback for the agent is a **random variable**. Therefore, our definition follows such conventions by defining regret based on the **expected accumulated rewards**, i.e., **the value function**. More importantly, defining a regret based on the value function is **NOT** proposed by us but quite common and well accepted in the RL literature, e.g. [5, 6, 7, 8], which all define regret based on the value function. Essentially, our contribution is not on proposing a brand new regret but on properly formulating and then solving our robust RL problem under the regret minimization framework.
>
> **Secondly, it's also possible to define regret using the exact reward as suggested by the reviewer, and it only changes the presentation of our results slightly.** Note that such a regret will become a random variable and the theoretical statements in our paper will be bounding the expectation of the regret. Therefore, defining regret based on the reward or its expectation are just different ways of presenting our results.

---

> ### Author Response · Authors · 2023-11-18
> **Response to Reviewer F1uH**
>
> > Q3: Also in equation 4.1, it seems that the regret depends on the attacker policy $\nu^t$ for each time $t$, where $\nu^t$ can be arbitrary. Is this regret well-defined without further confining $\nu^t$?
>
> **It is common to omit dependency of $\text{Regret}(T)$ on specific attackers for notational convenience since regret upper bound of interest for our paper and literature of online learning or adversarial bandit is for any $\\{\nu^t\\}_{t\in[T]}$.** It is true that regret depends on the specific attacker policy $\\{\nu^t\\}\_{t\in[T]}$, and notation for $\text{Regret}(T)$ could be really $\text{Regret}(T, \\{\nu^t\\}\_{t\in[T]})$ as suggested. However, in the literature of adversarial bandit and online learning, it is also common to just write $\text{Regret}(T)$ since all the regret guarantee of interest will be for arbitrary reward/loss sequences of the bandit's arms. For example, **such a convention to just write $\text{Regret}(T)$ also appear on Page 1 of [9], Equation 1.1 of [10], Figure 1. of [11], etc**, while regret does depend on the reward/loss sequence of the bandit's arms.  Similarly, when presenting the regret bound in our paper, for e.g., Proposition 4.5 & 4.7, it means the regret bound holds **for any unknown $\\{\nu^{t}\\}\_{t\in[T]}$**. We have made it clear in our revision.
>
> Finally, it is such a type of regret guarantee against any unknown attacker $\\{\nu^{t}\\}\_{t\in[T]}$ that ensures our approach can handle BOTH worst-case and non-worst-case attackers.
>
> ---
> > Q4: In the paragraph below equation 4.1, the term $\pi^tt\in[T]$ seems like typo. Should it be $\{\pi_t: t\in[T]\}$ instead?
>
> Yes, we thank the reviewer for pointing out the typo. We have revised our draft accordingly.
>
> ---
> > Q5: Another weakness is pointed out by authors in Theorem 4.11 that suggests the size of $\tilde{\Pi}$ can be very large in some cases that is exponential in time horizon $H$.
>
>
> 1. **Although the case requiring a large $\tilde{\Pi}$ is theoretically possible, it is an extreme case not taking the properties or structures of common practical applications into consideration**. The problem we constructed to prove Theorem 4.11 requires the attacker to perturb a state to another state with highly different reward functions. However, practical problems often pose certain constraints on the attacker, like bounded $l_p$ norm, and the reward function can be lipschitz. Thus, such a problem may not appear in practical applications.
>
> 2. **According to experiments, it is highly possible that the contrived problem instance we delicately designed in Theorem 4.11 will not appear**. With experiments on all environments (Section 4), small $|\tilde{\Pi}|$ within $5$ suffices for strong empirical performance. This further validates the practicality of our framework and the contrived problem may not happen.
>
> 3. **For practical infinite-horizon problems with a universal constant discount factor, the exponential dependency does not appear anymore**. As in our response to Q1, for infinite-horizon problems, we can construct a finite-horizon problem $H=\mathcal{O}(\frac{\log 1/\epsilon}{1-\gamma})$, where $\epsilon$ is the approximation accuracy. Therefore, if $\gamma$ is set as a universal constant as in most empirical implementations, dependency like $2^{H}$ will become $2^{\mathcal{O}(\frac{\log 1/\epsilon}{1-\gamma})}=\text{poly}(\frac{1}{\epsilon})$, where the exponential dependency will disappear. Therefore, in such cases, our hardness result may not hold anymore either.
>
> ---
> ### [2/2] Response to questions
> ---
> > Q6: Based on Theorem 4.11, will the method be very likely to fail for infinite horizon MDP?
>
> As in our response to Q1, our approach straightforwardly handles the infinite horizon MDP using a simple truncation. The approximation only has $H = \mathcal{O}(\frac{\log 1/\epsilon}{1-\gamma})$ steps, where $\epsilon$ is the approximation accuracy. Meanwhile, such a theoretical reduction is also compatible with empirical implementations, where even for infinite-horizon problems, one usually just samples finite steps. In conclusion, our method readily applies to infinite-horizon MDP.
>
> ---
>
> We greatly appreciate Reviewer F1uH's valuable feedback and constructive suggestions. We are happy to answer any further questions.

---

> > ### Author Response · Authors · 2023-11-18
> > **Response to Reviewer F1uH**
> >
> > [1] Zhang, Huan, et al. "Robust reinforcement learning on state observations with learned optimal adversary." arXiv preprint arXiv:2101.08452 (2021).
> >
> > [2] Sun, Yanchao, et al. "Who is the strongest enemy? towards optimal and efficient evasion attacks in deep rl." arXiv preprint arXiv:2106.05087 (2021).
> >
> > [3] Daskalakis, Constantinos, Noah Golowich, and Kaiqing Zhang. "The complexity of markov equilibrium in stochastic games." The Thirty Sixth Annual Conference on Learning Theory. PMLR, 2023.
> >
> > [4] Lattimore, Tor, and Csaba Szepesvári. Bandit algorithms. Cambridge University Press, 2020.
> >
> > [5] Azar, Mohammad Gheshlaghi, Ian Osband, and Rémi Munos. "Minimax regret bounds for reinforcement learning." International Conference on Machine Learning. PMLR, 2017.
> >
> > [6] Jin, Chi, et al. "Is Q-learning provably efficient?." Advances in neural information processing systems 31 (2018).
> >
> > [7] Liu, Qinghua, Yuanhao Wang, and Chi Jin. "Learning markov games with adversarial opponents: Efficient algorithms and fundamental limits." International Conference on Machine Learning. PMLR, 2022.
> >
> > [8] Dai, Yan, Haipeng Luo, and Liyu Chen. "Follow-the-perturbed-leader for adversarial markov decision processes with bandit feedback." Advances in Neural Information Processing Systems 35 (2022): 11437-11449.
> >
> > [9] Neu, Gergely. "Explore no more: Improved high-probability regret bounds for non-stochastic bandits." Advances in Neural Information Processing Systems 28 (2015).
> >
> > [10] Bubeck, Sébastien, and Nicolo Cesa-Bianchi. "Regret analysis of stochastic and nonstochastic multi-armed bandit problems." Foundations and Trends® in Machine Learning 5.1 (2012): 1-122.
> >
> > [11] Bubeck, Sébastien, and Aleksandrs Slivkins. "The best of both worlds: Stochastic and adversarial bandits." Conference on Learning Theory. JMLR Workshop and Conference Proceedings, 2012.
> >
> > Paper6801 Authors

---

> ### Author Response · Authors · 2023-11-20
> **Does our response address your concerns?**
>
> Dear reviewer F1uH,
>
> As the stage of the review discussion is ending soon, we would like to kindly ask you to review our revised paper as well as our response and consider making adjustments to the scores.
>
> Besides correcting all typos according to your valuable suggestions, we would like to point out that:
>
> (a)  We  have given a detailed explanation to show that our approach **does readily handle the setup of infinite-horizon MDP** and a comparison between **our regret notion and "the commonly used reward-based regret"** in the first response, and
>
> (b) We have offered a comprehensive analysis in the second response to show that exponential dependence on time horizon $H$ happens rarely.
>
> Please let us know if there are any other questions. We would appreciate the opportunity to engage further if needed.
>
> Best regards,
>
> Paper6801 Authors

---

> > ### Comment · Reviewer_F1uH · 2023-11-22
> > **Response to authors**
> >
> > Thanks much for providing detailed clarification. I think all my previous questions have been addressed.
> > Noting the theoretical contribution, I would be happy to change my score to 6, given that there are standing issues regarding the experiments as raised by other reviewers.

---

> ### Author Response · Authors · 2023-11-22
> **Thanks for your reply!**
>
> We are happy to hear that your previous questions have been addressed! Meanwhile, for the issues regarding the experiments, we have made the following efforts:
> - **Understanding the improvement on worst-case attacks**: We have provided experimental justifications **in Appendix E.4**, where existing approach explicitly trained on worst-case attackers suffers from algorithmic inefficiencies and instabilities.
> - **Understanding effects of our novel iterative discovery v.s. just using online feedback:** We have provided another experimental study **in the additional response to Reviewer M1op and our Appendix E.5**, where we create a new, stronger baseline that also utilizes the online feedback. We can see even compared with this, our approaches are consistently better, justifying our key contribution of iterative discovery for non-dominated policies.
>
> If there are any other questions from the reviewer, we are rather happy to address. We believe your comments will improve our paper greatly!
>
> Paper6801 Authors

---

### Author Response · Authors · 2023-11-18
**General response**

We thank all reviewers for their valuable feedback and insightful questions. We are encouraged that the reviewers find our approach for RL robustness against non-worst-case attack interesting (F1uH, M1op, t6xM, xB8p), and recognize both theoretical (M1op, t6xM, xB8p) and experimental (M1op, t6xM, xB8p) results of our algorithms. We believe we have addressed all reviewer's concerns in the separate responses. Below we summarize our responses and clarify some common confusion.

---
## Common concerns

### The notion of regret for online adaptation

Some reviewers doubt that our notion of regret may be not appealing (F1uH) or too weak (M1op). We would like to clarify as follows.

- **Our regret definition is standard and meaningful.** Its definition follows the most standard definition for RL problems, which is indeed based on the value function and compares with the best fixed policy in hindsight. It is well accepted by a series of previous works for regret minimization in RL as we mentioned in the separate response. **Our main contribution here is NOT proposing a novel notion but reformulating and solving robust RL problems within the regret minimization framework**.
- **Our framework is general and flexible for other variants of the standard regret.** Our method can also handle other variants of regret, for e.g. dynamic regret by replacing Algorithm 1 with other bandit algorithms for the corresponding notions.


### The improved performance of our method under worst-case attacks

The empirical results in our submission show that our method obtains better performance against worst-case perturbations (PA-AD) than the baselines explicitly trained against worst-case attacks. This may seem to be counterintuitive and cause confusion to some reviewers (M1op, xB8p).

- We think it is because our method, by **explicitly encouraging the new policy to behave differently to the discovered policies**, is less likely to be stuck at a single local optimum. For the baselines only aiming to maximize the worst-case robustness, they could suffer from converging to such local optima easily.
- To further verify our hypothesis, we compare the medium reward among **$21$ victim agents** trained with the same set of hyperparameters (in Table 1) with the best one among these $21$ agents.

| Environments  | Model       | PA-AD (against medium victim) | PA-AD (against best victim) |
| -----------   | ----------- | -------------- | --------------- |
| Hopper        | PA-ATLA-PPO | $2521\pm325$   | $3129\pm316$    |
|               | WocaR-PPO   | $2579\pm229$   | $3284\pm193$    |
|               | Ours        | $2896\pm723$   | $3057\pm809$    |
| Walker2d      | PA-ATLA-PPO | $2248\pm131$   | $3561\pm357$    |
|               | WocaR-PPO   | $2722\pm173$   | $4239\pm295$    |
|               | Ours        | $5803\pm857$   | $6052\pm944$    |
| Halfcheetah   | PA-ATLA-PPO | $3840\pm273$   | $4260\pm193$    |
|               | WocaR-PPO   | $4269\pm172$   | $4579\pm368$    |
|               | Ours        | $4411\pm718$   | $4533\pm692$    |
| Ant           | PA-ATLA-PPO | $2986\pm364$   | $3529\pm782$    |
|               | WocaR-PPO   | $3164\pm163$   | $4273\pm530$    |
|               | Ours        | $4312\pm281$   | $4406\pm329$    |

- **We notice that baselines like PA-ATLA-PPO and WocaR-PPO can match our high performance in terms of the best run, while the median one is low by a large margin.** This shows baseline methods can achieve high worst-case robustness occasionally as its objective encourages so explicitly but not reliably. In contrast, our methods are much more stable.

---
## Paper Updates

(All updates in our paper are highlighted in purple.)

- **Minor typos.** We corrected all minor typos mentioned by our reviewers in the revision.
- **Detailed explanations for $\text{Regret}(T)$.** To address the confusion of Reviewer F1uH, we use a footnote to clarify the dependency of $\\{\nu^t\\}\_{t\in[T]}$ omitted for notation convenience.
- **Additional experiments.** We add the ablation study on the worst-case robustness in Appendix E.4 to address the reviewers' confusion (M1op, xB8p). The supplementary results are shown in Table 2 in Supplementary.

---
## Summary of Novelty and Contributions
We would like to reiterate the following key points:
- **A novel formulation for robust RL.** We investigate both training and online adaptation phases under the prevailing state-adversarial attack model.
- **Algorithms for both two phases.** We propose two algorithms for online adaptation and training respectively. We also show theoretically that our algorithms are effective.
- **Empirical investigations.** We validate the effectiveness of the method on Mujoco, demonstrating both improved natural performance and robustness, as well as adaptability against unknown and dynamic attacks.

---

### Meta-Review · Area_Chair_apmp · 2023-12-13

**Metareview:**

This paper presents an RL policy learning approach for enhanced robustness across various attack scenarios. The method involves dynamically selecting policies from a predefined set, minimizing regret at each step. The set is constructed during training to be minimal while ensuring minimal performance gaps compared to the full set, achieved through identifying non-dominated policies. Theoretical guarantees are provided on problem hardness, optimality gaps, regret bounds, and worst-case set size. The approach outperforms baselines in Mujoco environments with continuous actions under no attack, non-worst-case attack, and worst-case attack conditions.

Strengths: The authors theoretically establish the difficulty of the problem, and present useful theoretical results regarding optimality gaps presented by the use of non-infinite policy classes, as well as the performance of their algorithm. The empirical performance is strong, with comparison on several environments against several baselines and attack strategies.

Weaknesses: As highlighted by Reviewer t6xM, "The discussion on adaptive attacks appears somewhat lacking—specifically, what are the potential implications if the adversary possesses more knowledge about the defender? The paper does not distinctly articulate the concept of adaptivity in this context. Nevertheless, we believe this aspect could be explored in future work, given the paper's introduction of innovative ideas." In the revised manuscript, it is recommended that the authors incorporate an adaptive attacker with superior knowledge (as mentioned in the authors' response) into the experiments. This adjustment will subject the proposed algorithm to a more adversarial setting, allowing for an assessment of its performance under such conditions.

**Justification For Why Not Higher Score:**

The paper's discussion on adaptive attacks is noted by Reviewer t6xM as somewhat lacking, specifically in addressing potential implications when the adversary possesses more knowledge about the defender.

**Justification For Why Not Lower Score:**

The experiment results look good.

---

### Decision · Program_Chairs · 2024-01-16

Accept (spotlight)